# Multi-scale and multi-context interpretable mapping of cell states across heterogeneous spatial samples

Patrick C. N. Martin[1], Wenqi Wang[2], Hyobin Kim [1], Henrietta Holze[2], Paul B. Fisher[3,4,5], Arturo P. Saavedra[6], Robert A. Winn[3,7,8], Esha Madan[3,4,5,7], Rajan Gogna [3,4,5,7,9] ✉ & Kyoung Jae Won [1,9] ✉

There is a growing demand for methods that can effectively align and compare spatial data in the absence of obvious visual correspondence. To address this challenge, we developed an interpretable cell mapping strategy based on solving a Linear Assignment Problem (LAP) where the total cost is computed by considering cells and their niches. We demonstrate that our approach outperforms other methods at capturing the spatial context of cells in synthetic and real data sets. The flexibility of our implementation enhances the interpretability of mapping and allows for accurate cell mapping across samples, technologies, resolutions, developmental and regenerative time. We show spatiotemporal decoupling of cells during development and patient level sub-populations in In Situ Mass Cytometry (IMC) cancer data sets. Our interpretable mapping approach facilitates systemic comparison and analysis of heterogeneous spatial data. We provide a flexible framework for researchers to tailor their analysis to the specific biological and research context.

From the initial discovery of cells through early microscopes in the 17th century, we have come a long way in our understanding of cellular organization[1]. Today, spatial transcriptomics (ST) has become an increasingly popular means of probing cellular organization in relation to cellular identity[2–6]. ST measures the gene expression profiles of cells as well as their location within a tissue. For clarity, we will refer to spatial spots, barcodes, or indices simply as cells. Taken together, ST methods - in their varying flavors – can highlight spatially resolved gene expression patterns[7–13]. They can demonstrate how tumors interact with their microenvironment and how disease can be understood from a spatial perspective[14–19]. They can provide insights into cellular communication through ligand-receptor dynamics as well as

cell-to-cell contact-triggered gene expression modulations[20–23]. Importantly, ST emphasized how cellular identity should be viewed through the lens of spatial context[23,24].

With the increasing availability and breadth of ST technologies, it comes as no surprise that they have become part of the biomedical research arsenal. It is still difficult and expensive to obtain high-quality spatial data, especially from human patients. Therefore, there is a need to maximize the use of the spatial data deposited in the public domain. However, datasets that are already available in a biological condition of interest might have been produced using different methodologies or technologies. Moreover, in a clinical setting where a researcher might be interested in following the evolution of cells and their interactions

[1]Department of Computational Biomedicine, Cedars-Sinai Medical Center, Hollywood, CA, USA. [2]Biotech Research and Innovation Centre (BRIC), University of Copenhagen, Copenhagen, Denmark. [3]Massey Comprehensive Cancer Center, Virginia Commonwealth University, Richmond, VA, USA. [4]VCU Institute of Molecular Medicine, VCU School of Medicine, Virginia Commonwealth University, Richmond, VA, USA. [5]Department of Cellular, Molecular and Genetic Medicine, VCU School of Medicine, Virginia Commonwealth University, Richmond, VA, USA. [6]Department of Dermatology, VCU School of Medicine, Virginia Commonwealth University, Richmond, VA, USA. [7]Department of Surgery, Virginia Commonwealth University School of Medicine, Richmond, VA, USA. [8]Division of Pulmonary Disease and Critical Care Medicine, Department of Internal Medicine, VCU School of Medicine, Virginia Commonwealth University, Richmond, VA, USA. [9]These authors jointly supervised this work: Rajan Gogna, Kyoung Jae Won. ✉e-mail: rajan.gogna@vcuhealth.org; KyoungJae.Won@cshs.org

across disease progression or as a response to treatment, it is difficult to expect tissues to share similar structures. And yet, we have seen the emergence of 3D spatial stacks[9,25,26] taken from adjacent tissues and, in parallel, a plethora of tools allowing the alignment of these tissue slices, including PASTE[27], GPSA[28], or STalign[29] have been developed. The assumption made by these tools is that adjacent sections share sufficient structural similarity to locally and differentially deform tissue sections to match the neighboring section. Importantly, these methods do not fold, break, or scramble the cells present in the tissue; rather, they displace them by maintaining their relative position to each other. But the irregular nature of tumors across patients and conditions restricts their use to highly structured tissues. As such, the question remains: how do we compare the spatial context of individual cells when tissue structures do not match across samples or where sequential sampling is simply impossible?

Mapping single-cell data to a spatial assay based on the transcriptome may suggest an alternative solution to align two or more ST assays based on their transcriptome[30,31]. For instance, Tangram uses a deep learning approach to match sc/snRNA-seq to their estimated spatial location by maximizing the spatial correlation between scRNA and spatial data[30]. CytoSpace solves a linear assignment problem (LAP) to match single cells to spatial locations by accounting for cell type proportions in low-resolution Visium spots[31]. In conjunction with single-cell labels, CytoSpace can also map to high-resolution spatial data. Along the same lines, methods designed to integrate single-cell data, such as Scanorama[32] can find the closest related cells in their integrated latent space. However, data mapping or integration without spatial and biological context can lose the information related to tissue or cellular microenvironment. Algorithms such as the Spatial-linked alignment tool (SLAT) provide a solution to spatial mapping by adopting a graph adversarial matching strategy[33]. Interestingly, SLAT provides an integrated latent space, making it a spatially aware data integration method. SLAT represents cells as neighborhood graphs and applies a graph convolutional neural network to generate its embeddings. This strategy can reduce both context flexibility and interpretability. While SLAT considers the spatial context of cells across samples, it has not been tested systematically for tissues whose structure is highly dissimilar. More importantly, current mapping strategies cannot be applied to a systemic, flexible, and interpretable comparison of hundreds of samples.

To address the challenge of mapping cells between structurally heterogeneous tissues, we developed an approach to map cells across 2 or more samples when structural heterogeneity impedes spatial alignment strategies. We map cells using a Linear Assignment Solver where the cost matrix is constructed by explicitly considering spatial or biological context. Specifically, we designed the cost matrix to minimize the pair-wise summation of multi-scale and multi-context spatial similarity matrices, including cell similarity, niche similarity, and spatial territory similarity (see Methods). Explicit use of context-specific matrices increases the interpretability of cell mapping results and provides sufficient flexibility to map samples across technology or collected over time. Importantly, our approach enables customized and interpretable analyses tailored to the specific research needs of individual researchers or datasets. We highlight the performance of our approach using synthetic datasets in a variety of scenarios with increasing complexity. We demonstrate how the performance observed in synthetic data can be translated to real biological datasets, including seqFISH mouse embryo[34], Stereo-seq mouse embryo[9], and Slide-seq V2 mouse hippocampus[8]. Vesalius accurately maps cells with their spatial context and outperforms other approaches. We demonstrate how our flexible framework can effectively map cells across datasets, technologies (seqFISH to Stereo-seq), and resolutions (VisiumHD to Visium). We also map cells across developmental and regenerative time in high-resolution Stereo-seq[9] data. This allowed us to highlight genes involved in brain development and regeneration,

which were only present in a limited subset of cells. Additionally, the cost of cellular mapping provides an interpretable and universally comparable metric to cluster in situ mass cytometry (IMC) breast cancer patient samples at the population level. The clustering results demonstrated how cellular context can recapitulate broad clinical metrics. Our cell mapping strategy grants us the ability to investigate biological differences between samples with unprecedented flexibility and interpretability.

## Results

### Mapping cells and their context across samples

With the increased availability of ST datasets, we aim to provide a flexible framework to compare spatial data by mapping a query dataset onto a reference dataset across conditions, technologies, and sections (Fig. 1a). To achieve this, we formulate the matching of cell pairs as a LAP where the goal is to match "a worker" to "a tasks" which minimizes the overall cost (Fig. 1b). To solve the assignment problem we used the Jonker-Volgenant algorithm[35]. In this instance, a "worker" is a cell from the query data, whereas the "task" is a cell from the reference data. The overall cost of mapping cells is defined by a pair-wise summation of score matrices across biological features (see "Methods" section - Fig. 1c). We account for transcriptional similarities between cells, their niches, their spatial tissue territory, cell type labels, and the cell type composition of their niche. In addition, we offer a straightforward method to add custom and context-specific matrices. The total cost matrix can be constructed from any combination of these biological features. For instance, while cell type labels can be used, they are not strictly required to map cells between samples. Our algorithm has been included in the Vesalius R package – a tool designed to retrieve spatial territories by applying image processing methodologies to spatial omics data[24] (Fig. 1d). The territory similarity matrix is computed using Vesalius's tissue territory detection capabilities[24]. The new and improved functionalities of Vesalius provide a framework for spatial mapping of cells across time, across technologies, and clustering hundreds of spatial samples.

### Benchmarking mapping performance in synthetic spatial data

To demonstrate the effectiveness of our mapping strategy, we elected to use synthetic spatial data since this will provide a robust ground truth (See "Methods" section). Specifically, we aimed to demonstrate our ability to accurately map cells across samples along with their spatial context. First, we generated 5 synthetic spatial regimes (*circle, layer, dropped, random one cell, random two cell*) that mimic the complexity of biological scenarios (Supplementary Figs. S1–S5). For each regime, we generated 12 samples where we aimed to map a query dataset onto a reference data for a total of 132 mapping events (excluding 12 self-mapping events) (Supplementary Figs. S1–S5).

Since Vesalius provides a flexible framework to construct the total mapping cost, we compared the performance of Vesalius across 14 different cost matrices in addition to two scRNA-to-spatial mapping tools (CytoSpace[31] and Tangram[31]), spatial alignment tools (PASTE[27] and GPSA[28]), and scRNA/Spatial data integration tools such as Scanorama[32] and SLAT[33]. The cost matrix combinations were built using cell similarity (f = feature), niche similarity (n = niche), territory similarity (t = territory), niche cell type composition (c = composition), cell type labels (y = cell type labels), or a combination of these metrics.

To measure the effectiveness of each tool in mapping cells across samples, we computed an Adjusted Rand Index (ARI)[36] between mapped cell labels. In parallel, we measured each tool's ability to capture spatial context by adding an "interaction" label to our datasets (see "Methods" section). This label concatenates the cell type labels of the k-nearest neighbors ($k = 6$) of a center cell. Using these labels, we computed a Jaccard index (JI) to ascertain how similar the neighborhoods are in terms of cell type composition.

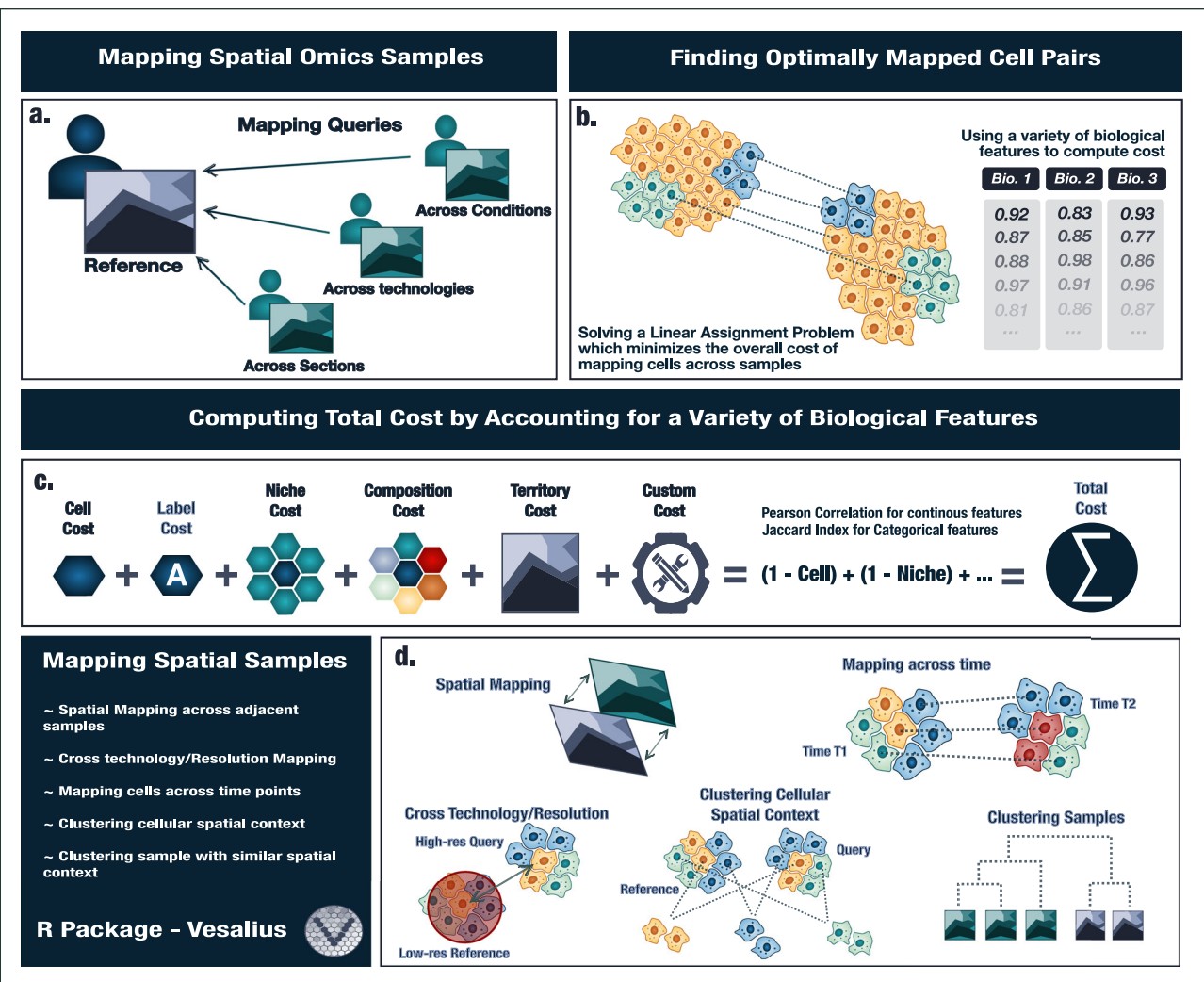

**Fig. 1 | Overview of Vesalius's cell state mapping strategy. a** Samples across conditions, technologies, and sections are likely to present a heterogeneous structure, and yet, to capitalize on the growing wealth of spatial omics data, it is crucial to accurately map cellular context across conditions. **b** To achieve cell mapping in heterogeneous spatial samples, Vesalius solves an LAP, which aims to minimize the overall cost. To account for spatial context during mapping, Vesalius leverages a multitude of biological features. **c** The total cost matrix can be constructed from a pair-wise summation of reciprocal similarity scores, including cell similarity, label similarity, niche similarity, composition similarity, territory similarity, and even custom similarity scores. **d** Vesalius maps cells across samples, across time, and across technologies. The total cost also provides a useful metric to define patient-level spatial similarities and allow for sample clustering. Source data are provided as a Source Data file.

Vesalius demonstrated performance satisfying both cell type score (ARI) and interaction score (JI) in these tests. When measuring the ability to recover the identity of mapped cells in the *circle* regime, for instance, Vesalius showed comparable performance with SLAT, Scanorama, and CytoSpace (Fig. 2a) in ARI. The perfect score of CytoSpace was expected, as it only maps cells of the same cell type. With its requirement of shared cell type labels between datasets, CytoSpace was incapable of mapping samples when not all cells were shared (121 were mapped out of 132). This was particularly evident in the *dropped* regime, where CytoSpace only mapped 7 samples out of 132 (see Supplementary Fig. S7). We show the mapping performance for all other regimes in Supplementary Figs. S6–S9. The performance of CytoSpace dropped dramatically when cell labels were removed (*CytoSpace_noLab* in Fig. 2, Supplementary Figs. S6–S9).

If we consider each tool's ability to map spatial context, we see that Vesalius outperforms all other tools when including the cellular context to construct the cost matrix. Unsurprisingly, including neighborhood composition strongly improved Vesalius's performance. It is of note that CytoSpace (with true labels) and Scanorama

excelled at finding the right cell type but failed at recovering context, falling behind Vesalius even when it did not include cell composition (e.g., *n, fn, nt* cost matrices – legend key shown in figure legend). We show pair-wise statistical comparison (Wilcoxon Rank Sum test- see methods) between each tool for ARI scores and JI scores in Fig. 2c, d, respectively.

We exemplify the performance of each tool by plotting a mapping event in the *circle* regime (Fig. 2e). We aimed to map cells in the query sample onto the reference datasets (panels in the top left of Fig. 2e). For clarity, we selected only 3 cost matrix combinations for Vesalius. Specifically, we selected *fncty, fnct,* and *fnt*. Our results indicate that Vesalius was able to effectively map cells from the query sample onto the reference sample while considering spatial context. We noticed that spatial alignment tools such as PASTE and GPSA failed to recover the structure of the reference sample from the query sample. The task of mapping cells across samples is a distinct task from spatial alignment, and the constraints applied to the samples by PASTE and GPSA are incapable of overcoming these differences. We show the results for all other synthetic datasets in Supplementary Figs. S6–S9.

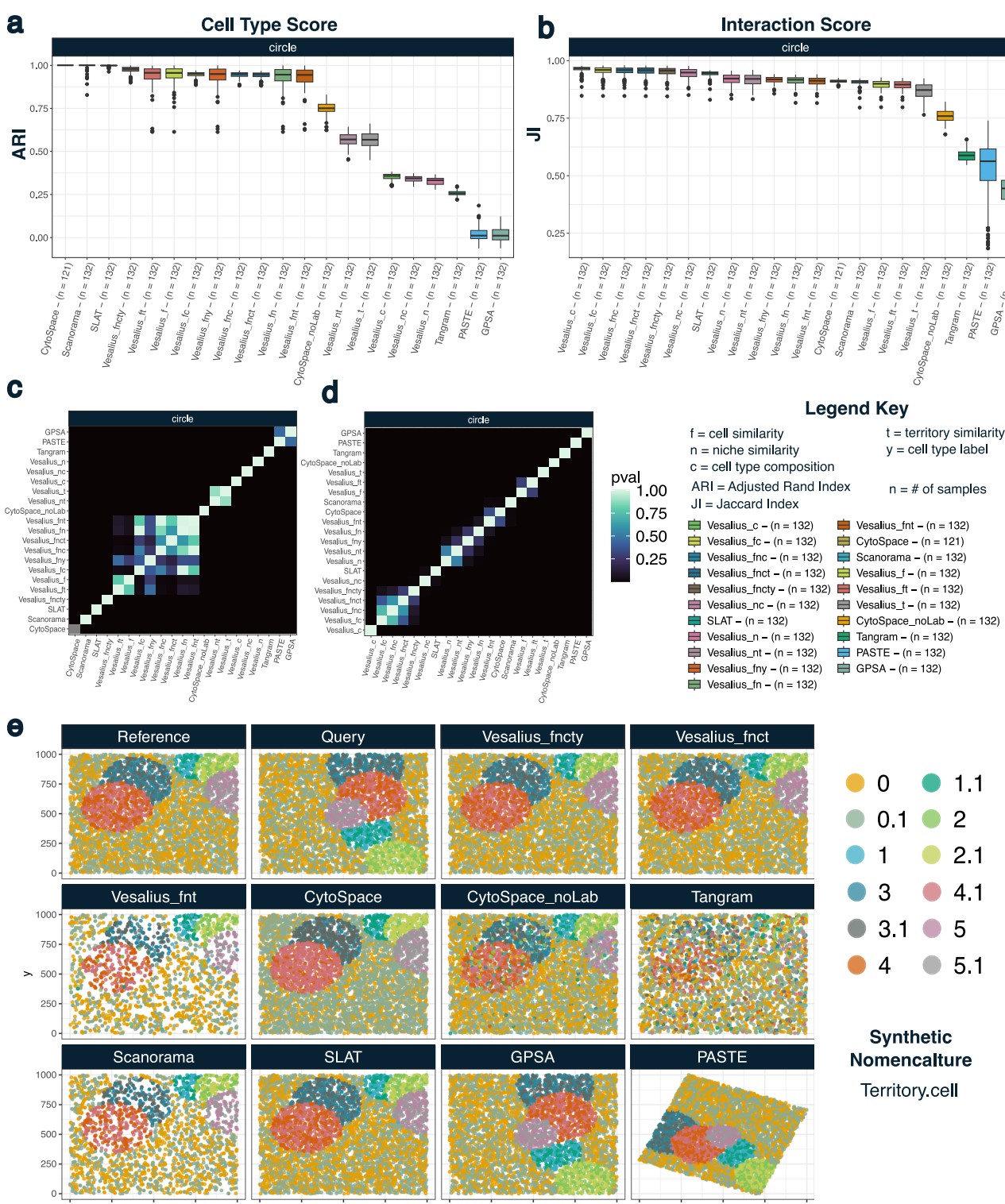

**Fig. 2 | Benchmarking mapping performance in synthetic spatial data – circle regime.** We provide a Legend Key on the figure describing the nomenclature of Vesalius cost matrices as well as the meaning of acronyms. **a** Performance comparison in recovering cell types across samples using an ARI in synthetic spatial data. We compared Vesalius using 14 different cost matrices to SLAT, CytoSpace, CytoSpace with no cell type labels (noLab), Tangram, Scanorama, PASTE, and GPSA. The number of samples (*n*) used in each box plot is included in the axis labels. The box shows the interquartile range (IQR = Q3 − Q1), with a line at the median, and whiskers extending to the most extreme values within 1.5× IQR. CytoSpace obtains a perfect score since it only maps cells from the same cell type. **b** JI of cell interactions. To account for the spatial context of cells, we computed the JI between cell type labels in the neighborhood of mapped cells (*k* = 6). **c** Heat map representing *p*-values after performing a two-sided Wilcoxon rank sum test between tools on ARI scores. **d** Heat map representing *p*-values after performing a two-sided Wilcoxon rank sum test between tools on JI scores. **e** Example mapping event in the circle regime. In the circle regime, there are 5 randomly sized and randomly placed territories in a background territory. Each circle can contain 2 different cell types. We show mapping events for 3 Vesalius cost matrices (*fncty, fnct, fnt*). Source data are provided as a Source Data file.

Finally, we highlighted computational performance (run time and memory usage) when increasing the number of cells from 100 to 200,000 (Supplementary Fig. S10) in a synthetic circle *regime* (See "Methods" section). For each tool, we computed the mean run time and peak RAM usage (Resident Set Size) across 3 runs for each number of cells. We also included a comparison of run time and memory usage across cost matrix combinations for 5000 and 10,000 cells.

## Benchmarking mapping performance in real spatial data

We further tested Vesalius using the same 14 cost matrices against other tools (excluding GPSA due to excessive memory requirements) in seqFISH mouse embryo[34], Stereo-seq mouse embryo[9], and Slide-seq V2 mouse hippocampus[8] (see "Methods" section). We found that the mapping by Vesalius with all available information (*fncty*) showed ARI larger than 0.75 for all datasets we tested. This was compared with other approaches, whose performance was substantially compromised in real datasets. Again, the perfect ARI score for CytoSpace was expected as it only maps to cells with the same cell type. When the cell type information is lost (CytoSpace_noLab), the ARI becomes less than 0.3 for all datasets. Due to the requirement of matching cell type, CytoSpace cannot run on the Stereo-seq dataset without removing label information (Supplementary Fig. S11). SLAT, which showed good performance for synthetic datasets, showed ARIs around 0.5 and even 0.25 for Slide-seq data. Regardless of the datasets, Vesalius showed the best performance when assessed with JI, especially when using cell composition (*c*). The JI score decreased when adding other configurations to compute the cost in Vesalius, while ARI, in general, is enhanced, indicating that Vesalius tries to find a balance between cells and spatial context depending on the configuration.

We show mapping examples using *fncty, fnct,* and *fnt*. The mapping examples of Vesalius also demonstrated a strong resemblance to the reference (Fig. 3c, f, and Supplementary Fig. S11c). While SLAT's mapping generally aligned with the reference, it exhibited misannotations in highly heterogeneous brain regions (Fig. 3f). SLAT and Scanorama still perform decently in mapping cells across seqFISH samples but fell behind in capturing the cellular context. In fact, Vesalius, using only cell and niche similarity, performs similarly to SLAT at capturing cellular context. Again, PASTE failed to recover the structure of the reference sample from the query sample as observed in the examples using synthetic data.

Finally, we further investigated the role played by each cost matrix in mapping cells across samples. We first computed a coefficient of variation (CV) on each cost matrix across these 3 biological samples (seqFISH, Stereo-seq, and Slide-seq V2). A higher CV would indicate a higher ability for a cost matrix to discriminate cells during the mapping. We also computed a Proportion of Contribution (POC), which measures the proportion of contribution of each cost matrix to the total cost used for mapping (Supplementary Fig. S12). A higher POC is associated with higher JI or correlation. Cell type labels and neighborhood composition showed a higher ability to discriminate cells during mapping, but participated much less in the total mapping cost. On the contrary, territory and niche similarity tended to have a low CV but contributed significantly to the total cost of mapping.

## Mapping cells across technologies and resolutions

More often than not, datasets of interest are generated using different technologies, and yet we would still like to mine insights from them. We tackled this challenge using our flexible mapping strategy. First, we mapped high-resolution Stereo-seq mouse embryo[9] (E9.5) onto image-based seqFISH[34] in the same tissue at a slightly earlier developmental stage (E8.75). While both datasets contain cell type labels, these differed greatly between datasets, as is often the case when using data from different sources. As such, we forwent the use of cell type labels, and we constructed the total cost from cell similarity and niche similarity only. Overall, the broad cell type labels matched well when

mapping across technologies (Fig. 4a). We show the mapping of cells to different cell labels in an alluvial plot (Fig. 4b). For instance, Brain and Notochord cells in Stereo-seq are mapped to Forebrain/Midbrain/Hindbrain and Spinal Cord in seqFISH (accuracy = 0.901). We show the cell mapping results for Stereo-seq to all 3 seqFISH sections (Supplementary Fig. S13) and performed the converse mapping - seqFISH mapped to Stereo-seq (Supplementary Fig. S14).

To test mapping across spatial resolutions, we mapped the VisiumHD mouse brain onto Visium of the same tissue (See "Data availability" section). We elected to use the 8 μm bins since this would strike a balance between having high-resolution and reducing noise at each spatial location. In this context, we reasoned that, given the difference in resolution, the most appropriate strategy was to only consider niche similarity and territory similarity. More specifically, for each cell, we defined the niche by using a distance radius around the center cell (see "Methods" section). We defined the radius as equal to the radius of a Visium spot to ensure that a Visium niche would only contain a single spot, while VisiumHD would contain all spots within that radius. We detected tissue territories in the reference Visium sample and the query VisiumHD sample (Fig. 4c) and recovered expected tissue structures such as the CA1 field, CA3 field, and the Dentate Gyrus. Since mapping high-resolution data onto low-resolution data signifies that multiple spots will be mapped to the same low-resolution spots, we added random noise (jitter) to the coordinate values to better visualize the mapping results. Alternatively, we omitted the jitter to create pseudo-Visium data where each mapped cell will be merged into a mini-bulk count data at the spot level (Fig. 4c – left panels). We discovered that we were recovering the expected tissue structures, demonstrating that we can effectively map cells across resolutions. Interestingly, we observed finer tissue territories in the mapped region of the Visium assay compared with the reference.

To further investigate the cell types' matching abilities between resolutions, we used RCTD[37] to estimate cell type proportions in both Visium and VisiumHD using the same reference scRNA-seq dataset[38]. We scaled cell type frequencies (see "Methods" section) to create a contingency matrix of cell type proportions. We performed a Chi-squared test to determine whether cell type proportions exhibited significant differences between Visium and mapped VisiumHD. The *p*-value distributions across all mapped cells show that nearly all mapped cells display similar cell type proportions (Fig. 4d). When we discretized the p-value, we observed only a few instances where cell type frequencies statistically differed. In addition, we computed a JI between cell types associated with Visium spots and the mapped cell identities from VisiumHD. Our results showed a reasonable correspondence in mapped cell types between Visium and VisiumHD (Fig. 4e). The distribution of Jaccard scores showed that the median Jaccard score was around 0.32 (Fig. 4e - right panel with 25th, 50th, and 75th percentiles shown by vertical orange, blue, and green lines). Taken together, our results indicated that while Vesalius occasionally missed cell types associated with Visium spots or VisiumHD cells, the cell type frequencies did not statistically differ.

## Mapping cells across time in brain regeneration and embryo development highlights spatiotemporal expression patterns among mapped cells

The STOmics database provides a wide variety of high-resolution Stereo-seq data collections. Most notable are the MOSTA[9] and ARTISTA[26] data collections, which provide spatial-temporal maps of murine embryo development and Axolotl brain regeneration. Using these collections, we applied our approach to mapping cells across developmental time. First, we focused our attention on brain regeneration with the aim of mapping cells forward in time. We selected 20 Days Post Injury (DPI) as the reference onto which we would map 15

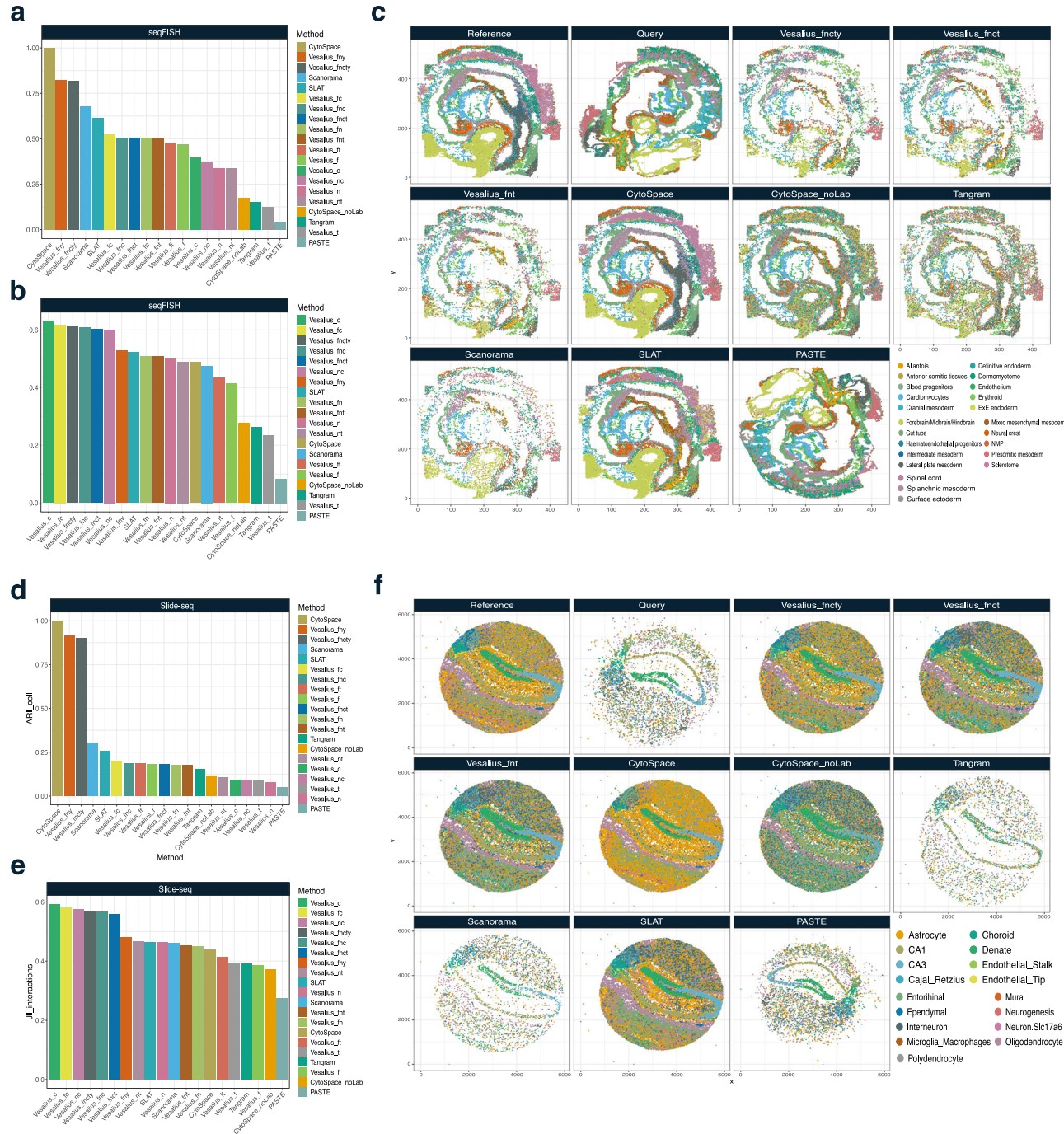

**Fig. 3 | Benchmarking mapping performance in real spatial data. a** ARI of mapped cell type labels between a reference seqFISH mouse embryo dataset and a query seqFISH mouse embryo dataset. We compared Vesalius using 14 different cost matrices to SLAT, CytoSpace, CytoSpace with no cell type labels (noLab), Tangram, Scanorama, and PASTE. In seqFISH data, the addition of cell type labels or composition of cellular neighborhoods improves Vesalius's ability to accurately map cells across samples, even outperforming SLAT and Scanorama. **b** JI of cell

interactions of the neighborhood ($k = 10$). **c** Example mapping event in seqFISH mouse embryo. We show mapping events for 3 Vesalius cost matrices (*fncty, fnct, fnt*). **d** ARI of mapped cell type labels between a reference Slide-seq V2 mouse hippocampus dataset and a query Slide-seq V2 mouse hippocampus dataset. **e** JI of cell interactions of the neighborhood ($k = 10$). **f** Example mapping event in Slide-seq V2 mouse hippocampus. We show mapping events for 3 Vesalius cost matrices (*fncty, fnct, fnt*). Source data are provided as a Source Data file.

DPI (Fig. 5a) with a cost matrix constructed from cell similarity, niche similarity, and territory similarity. Since cell type labels will differ between time points, we reasoned that these 3 cost matrices represented the most appropriate combination of metrics. We observed that cells are accurately mapped in the 20DPI brain (right side of the spatial assay). Ependymoglial cells (ECG) – equivalent to neural stem cells – formed a mass of reactive ECG cells (reaECG) where

regenerative intermediate progenitor cells (rIPC3 and rIPC4 cells) will form at a later stage (Highlighted in Fig. 5a by dark blue circle).

We created an integrated image stack (see "Methods" section) upon which we applied image processing and isolated common territories between reference and query datasets (Fig. 5b). The left assay represents the mapped cells while the right represents the reference cells. The temporal alignment provided a unique opportunity to

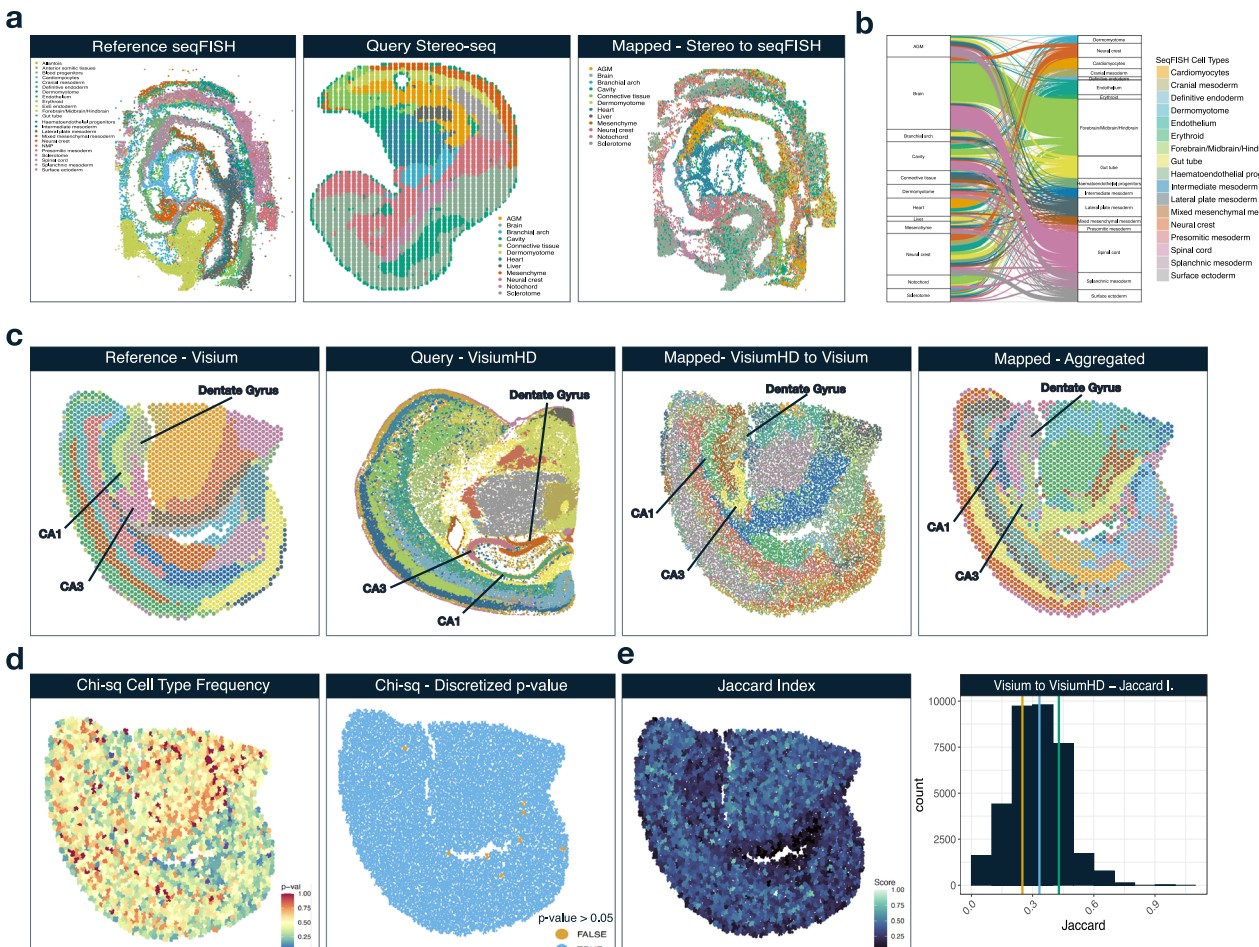

**Fig. 4 | Cell Mapping across technologies and resolutions. a** Mapping of Stereo-seq mouse embryo cells onto seqFISH mouse embryo cells shows that Vesalius recovers expected tissue structures across technologies. **b** Alluvial plot demonstrating where each mapped cell type in the query (Stereo-seq) maps to in the reference (seqFISH). **c** Mapping VisiumHD mouse brain onto Visium mouse brain with the tissue territories detected by Vesalius in the reference, the query, as well as the mapped cells with a jitter added to the coordinates. We recover the overall structure of the mouse brain, including the Dentate gyrus and the Cornus Amonis fields (CA1 & CA2 – annotated on figure). **d** The mapped p-values after performing a Chi-squared test between cell type frequencies in Visium (after cell type deconvolution) and the mapped cell types taken from VisiumHD. A discretized p-value map shows that only a few spots exhibit a statistically different cell type frequency between spots and mapped cells (p-value < 0.05). **e** JI of mapped cell labels between Visium and VisiumHD. 50% of cells have a JI score above 0.32. The orange vertical line represents the 25th percentile. The light blue line represents the 50th percentile. The green line represents the 75th percentile. Source data are provided as a Source Data file.

identify differentially expressed genes, focusing on associated cell pairs. We computed differentially expressed genes between reference and mapped cells at the injury zone (territory 8 – dark blue) to uncover which genes were involved in the regeneration process (two-sided Wilcoxon rank sum test – p-value < 0.05 – FDR adjusted). Out of the 123 differentially expressed genes, we found 2 notable genes *AMEX60DD048805* (*EDNRB*) and *AMEX60DD022398* (*ARPP19*). The former – homologous to the EDNRB human gene involved in vasculature and cell proliferation – was expressed in a subset of vascular leptomeningeal cells (VLMC) in the outer layers of the brain (Fig. 5c). The latter – a gene involved in the cell cycle, was expressed in a specialized set of ECG cells, srfpECG, and wtnEGC. The full list of differentially expressed genes is available in Supplementary Data 1.

We applied a similar process of mapping forward in time in the mouse embryo. As an example, we selected developmental stage E12.5 as the reference data and E11.5 as the query data. The total cost matrix was constructed using cell similarity, niche similarity, and territory similarity using the same reasoning as for the brain regeneration data. Once again, our approach showed a remarkable ability to map cells across space and time (Fig. 5d). We aimed to explore how cell populations preferentially mapped to the same set of cells in the reference data. To achieve this, we performed hierarchical clustering (see "Methods" section) using cell similarity in brain cells taken from the query data and identified 5 clusters (Fig. 5e). We observed spatial distinct structures in the mapped cells: for instance, cluster 2 and 4 highlight cells that were mapped at the interface of these brain regions (Fig. 5e). We computed which genes were enriched in each cluster (two-sided Wilcoxon rank sum test – p-value < 0.05 – FDR adjusted). For instance, *Hes5*, *Stnm2*, and *Cacng4* were differentially expressed in clusters 1, 3, and 4, respectively (Supplementary Data 2). Spatial gene expression patterns for each cluster indicated that our mapping strategy recovered expected gene expression patterns across space and time (Fig. 5f). Intriguingly, we discovered examples of spatio-temporal decoupling (Supplementary Data 3). For instance, *Crabp1* (Cellular Retinoic Acid Binding Protein 1), a gene involved in stem cell proliferation and differentiation[39,40], shows high expression in cluster 3 of the query cells but is not expressed in the reference data. To demonstrate that our mapping strategy was able to map cells across highly heterogeneous tissues such as cancer, we mapped cells from prostate cancer tumor samples using Slide-seq V2 technology. Using the metadata provided by the authors, we mapped cells across

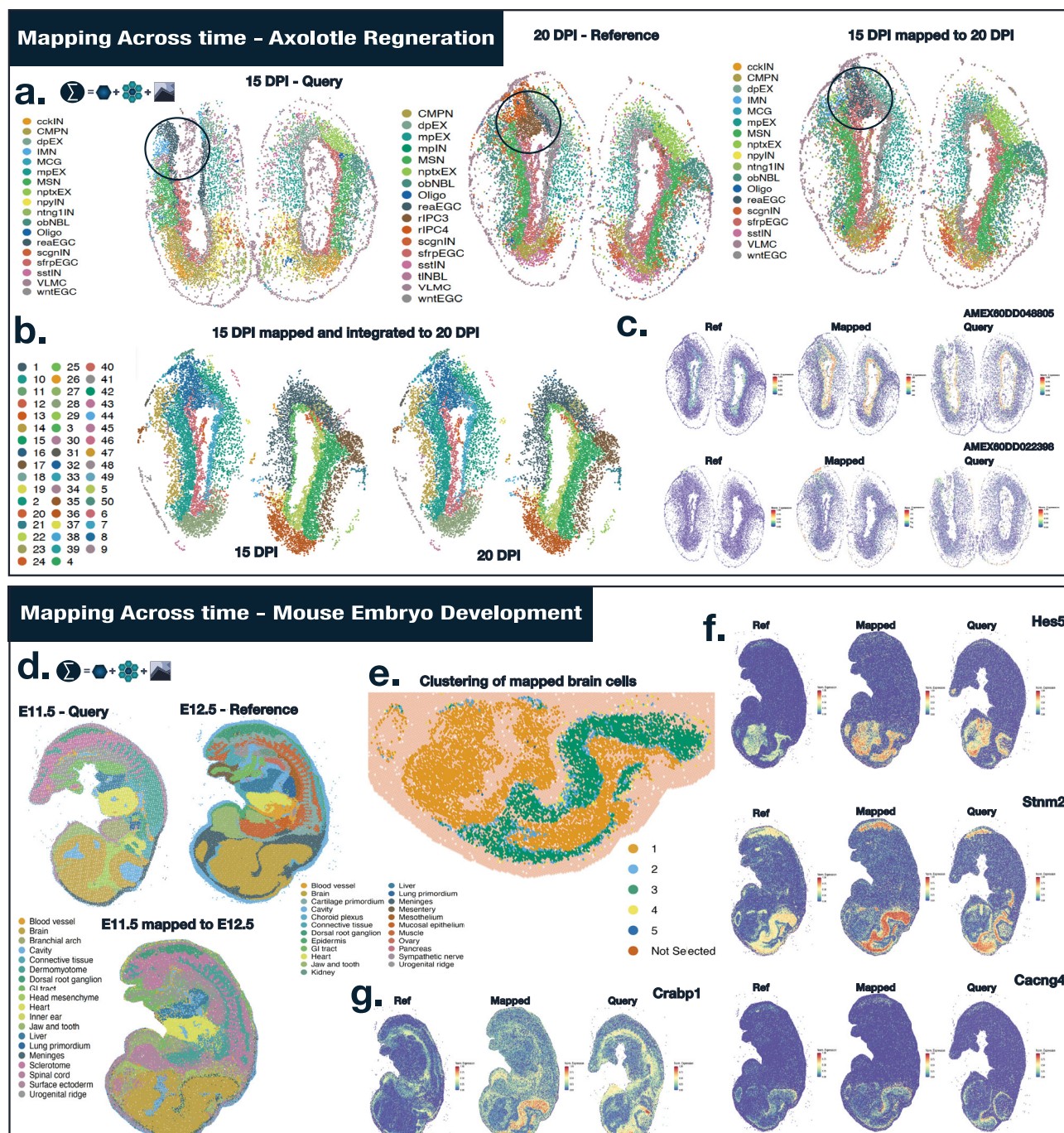

**Fig. 5 | Cell Mapping across time. a** Mapping Stereo-seq Axolotl brain regeneration datasets (ARTISTA) at 15 days post injury (DPI) to 20DPI using cell, niche, and territory similarity. Blue circle highlights ependymoglial cells and their transition to regenerative intermediate progenitor cells. **b** Integrated tissue territories of mapped cells and reference cells where tissue territories are recovered using a shared latent space. For visualization purposes, we separate both samples. **c** Two differentially expressed genes (two-sided Wilcoxon rank sum test – p-value < 0.05 – FDR adjusted) at the zone of injury (territory 8 – dark blue): AMEX60DD048805 and AMEX60DD022398 show expression patterns limited to a small subset of cells involved in cell differentiation and proliferation necessary for wound healing. **d** Mapping Stereo-seq Mouse embryo (MOSTA) development at stage E11.5 to stage E12.5 using cell, niche, and territory similarity. **e** Clustering of query brain cells

mapped onto the reference brain cells. Brain cells are demarcated into sub-territories, including cells that lie at the interface between brain regions (cluster 2) and the interface between the brain and other tissues (clusters 4). **f** Differential gene expression (two-sided Wilcoxon rank sum test – p-value < 0.05 – FDR adjusted) between clusters exemplified by *Hes5*, *Stnm2*, and *Cancng4* as genes highlighting the different spatial sub-structures present in the mouse brain and the accurate mapping of gene expression patterns across time. **g** *Crabp1* – a gene involved in the regulation of stem cell differentiation – shows a spatial-temporal decoupling behavior where the expression shows a spatial demarcation at the earlier developmental stage (E11.5), which is then extinguished at the following developmental stage. Source data are provided as a Source Data file.

samples using cell, niche, territory, and cell type labels to construct our cost matrices. We show a mapping example in Supplementary Fig. S15. Our results indicate that even in highly heterogeneous cancer samples, Vesalius could map cells across samples.

### Mapping and Sample stratification through cellular context mapping in cancer data

We further applied our strategy to align hundreds of spatial samples. We randomly sampled 100 patients from Breast Cancer IMC data containing a total of 37 protein markers[41]. We conducted an initial filtering step to ensure that all datasets contained clinical-level metrics such as ER Status (Positive or Negative), ERBB2 positive, cancer grade, death, cancer subtype (PAM50), and years to outcome. We then selected samples that contained at least 1000 cells. From the remaining datasets, we used ER Status to select patient sub-populations since this metric enables balance and interpretability in our patient sampling procedure. We then mapped each sample to all other samples (excluding self-mapping) using a cost matrix generated from cell similarities, niche similarities, territory similarities, cell labels, and niche composition. Here, we aimed to leverage all available information. From each mapping event, we extracted the average cost and mapping scores to define overall sample mapping performance.

We selected the best mapping pairs based on the lowest overall cost (Fig. 6a). Across all pairs, cell similarity, niche similarity, and territory similarity exhibited the highest average mapping scores. Niche composition had the lowest overall score. A large fraction of the best-matching pairs also showed high correspondence with clinical metrics, with 80% having at least 3 out of 5 metrics correctly matched (Fig. 6b). ERBB2_pos was the most well-predicted feature for our cost function, followed by ER Status. The least well predicted was cancer subtype (PAM50) with 49 out of 100 correctly predicted. However, this metric contains 5 sub-categories (Basal, HER2, Luminal A, Luminal B, Normal-like), making our predictions well above a random baseline (20). The majority of samples showed a low difference in years to outcome between query and reference samples.

We further performed hierarchical clustering of the ER-negative patients ($n = 50$) using our total cost as a measure of distance between samples (Fig. 6c). In addition to cost, we show the average performance associated with each cost metric. In total, we found 14 ER-negative sub-populations. Some clusters were driven by cell composition and cell type labeling (clusters 10, 9, and 4). We did not observe as strong a difference in mapping scores related to cell similarity, niche similarity, and territory similarity in these clusters. We checked which clinical metrics were associated with these clusters (Fig. 6d). Clusters 10 and 9 were characterized by Basal and HER2 cancer subtypes, respectively. Our results suggest that cell type labels of neighboring cells can dictate clinical metrics. Clustering results of ER-positive patients are shown in Supplementary Figs. S16 and S17. Labels and clustering results are also available in Supplementary Data 4.

## Discussion

With the growing volume of spatial data, the necessity to map and compare cells across samples has become increasingly crucial. To achieve this goal, it is inevitable that we need to reconcile spatial datasets that are dissimilar in their spatial structure, collected using different technologies, different capture resolutions, or might have been collected under different conditions or time points. More importantly, we need to achieve this aim in such a way that flexibility, interpretability, and understanding of the underlying biology are possible. To address the challenge of mapping cell states across heterogeneous samples, we developed a new approach within the framework of Vesalius – a spatial omics toolkit.

Mapping cells across heterogeneous samples is a distinct task from the spatial alignment of spatial samples. Tools such as PASTE[27], GPSA[28], or STalign[29] make an assumption of structural rigidity, which is expected in adjacent tissue samples. While they can locally and differentially deform a query sample to match a target sample, the query sample cannot be broken, folded, or scrambled. However, samples taken from a tumor (even if it is the same tumor) are going to be structurally heterogeneous, and this heterogeneity will be further amplified when sampling across patients. We demonstrated in our benchmarking (synthetic and real data) that spatial alignment tools (PASTE and GPSA) are incapable of mapping cells between samples in a way that will recover the structure of morphologically complex tissues.

The many-to-many cell matching framework is similar to scRNA-seq mapping tools such as CytoSpace[31] and Tangram[31]. Neither tool, however, includes spatial information to map cells across samples. Alternatively, Scanorama[32] and SLAT[33] provide single-cell data integration methods, with SLAT utilizing spatial neighborhoods to construct its integrated latent space. However, the inclusion of neighborhood information alone is not always sufficient to accurately map cells across samples. In fact, capitalizing on all available information provides more accurate spatial mapping outcomes (Figs. 2 and 3).

In our benchmarking across synthetic and real datasets, we demonstrated how the inclusion multi-scale and multi-context information improves the mapping of cellular spatial context. We observed that tools such as SLAT and Scanorama perform well at accurately mapping cells with the same cell type across synthetic samples (Fig. 2a, Supplementary Figs. S6–S9). However, they did not capture spatial context with sufficient accuracy to map the cellular and spatial context of cells (Fig. 2b). Vesalius, on the other hand, could exploit all available information (including cell type composition of niches and cell type labels) to accurately map cells and their spatial context. It is of note that Vesalius showed high performance when dealing with more complicated real data where multiple cell types are co-localized, indicating that Vesalius utilizes spatial context effectively for mapping. In contrast, the high performance of SLAT and Scanorama was not reproduced in real biological data.

CytoSpace proved to be an interesting case study on the over-reliance on certain modalities. CytoSpace consistently obtained perfect scores at mapping cell identities across samples in both synthetic and real data. However, this is due to the mechanism by which CytoSpace utilizes cell type labels: Only cells with the same cell type label will be mapped to each other. In fact, when cell labels are not shared or removed CytoSpace either fails to run or performs poorly. Even with cell type labels, it does not capture the spatial context of cells (Fig. 2b, Fig. 3b/e). While Vesalius can also capitalize on cell type labels, it does so with much more flexibility. Cell type labels constitute only part of the total cost matrix, and if a cell type label does not match, but the other scores (cell, niche, or territory similarity, for example) are high, then this mapping event might still occur. This mechanism allows Vesalius to utilize cell labels while still accounting for the cell state continuum and compensating for inaccurate cell type labels.

The flexibility with which Vesalius constructs its cost matrices encouraged us to explore how we can employ this mechanism across a variety of analysis scenarios. First, we aimed at mapping cells across technologies as distinct as seqFISH and Stereo-seq in mouse embryos (Fig. 4a/b). Despite the technical differences, Vesalius showed himself to be particularly adept at mapping cells across technologies by utilizing the available information (cell and niche similarity). We employed this flexibility with great success when mapping cells across spatial resolutions. By using niches defined by a spatial radius and large-scale territory information, we were able to recover expected cell type composition between Visium and VisiumHD mouse brain data (Fig. 4c–e). Second, we mapped cells across time in Axolotl brain regeneration and murine embryonic development, where we were able to recapitulate the expected tissue structures despite only using cell, niche, and territory similarities to construct our total cost (Fig. 5). Our integrated territory approach allowed us to underline genes being

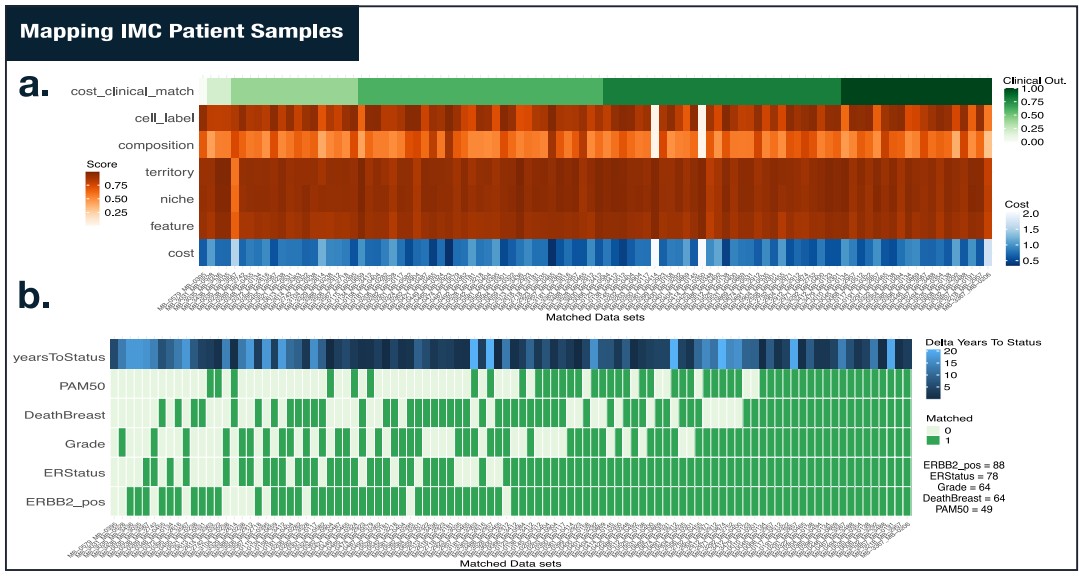

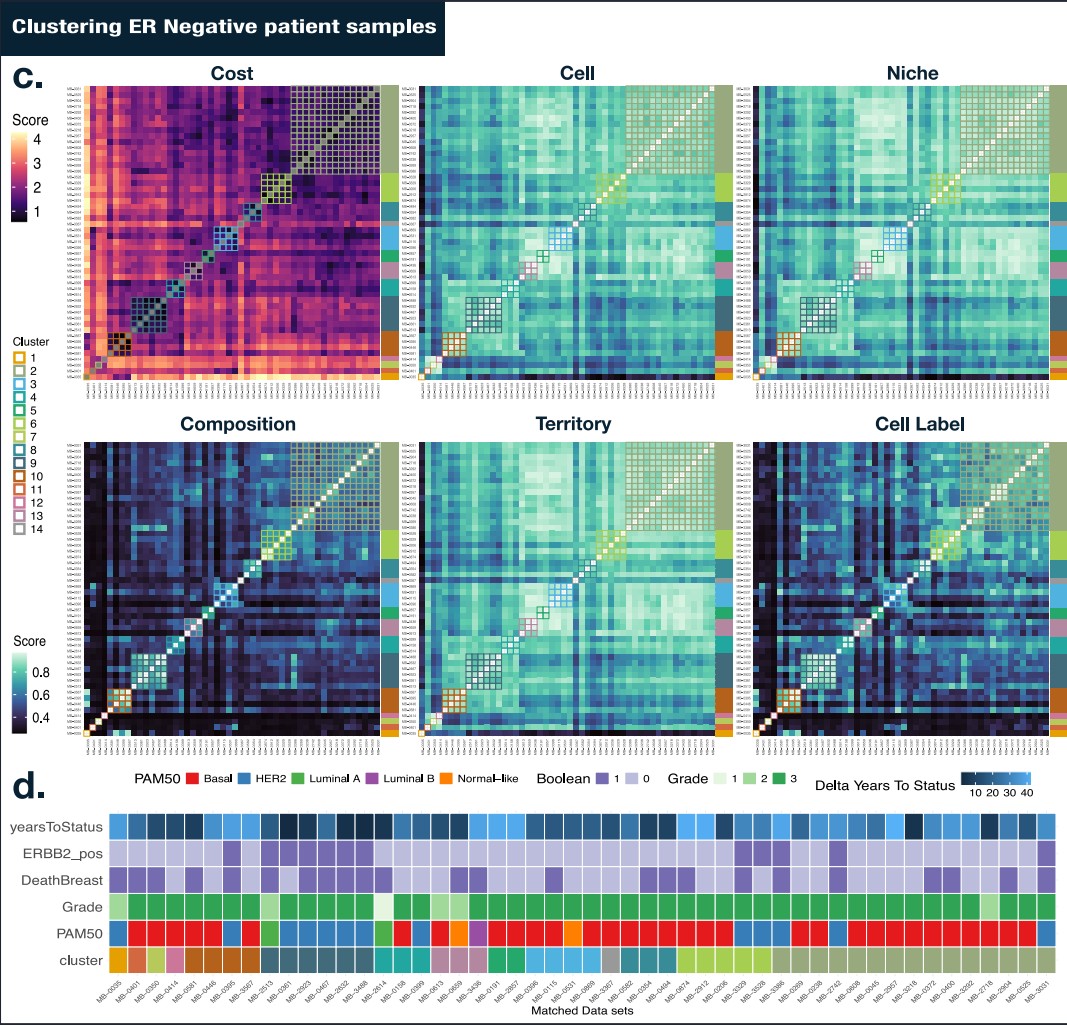

activated at the zone of injury. Brain cells in the mouse embryo demonstrated spatial sub-populations, which led to the discovery of spatial-temporal decoupling events related to stem cell differentiation.

In addition to its flexible cost construction mechanism, Vesalius uses simple and interpretable scores. A Pearson's correlation coefficient or a JI represents universally interpretable measures of how similar or dissimilar 2 datasets are. In contrast, distances within a

spatially aware integrated latent space computed by deep learning methodologies (such as SLAT) are difficult to interpret. They do not highlight which aspects of a spatial dataset can be used to discriminate samples. To this effect, we used Vesalius's interpretability to our advantage by clustering spatial samples. We applied this sample clustering approach across 100 breast cancer IMC samples and found remarkable concordance with clinical metrics such as ER status and

**Fig. 6 | Mapping and clustering of IMC samples. a** Scores of the best-matching patient pairs across 100 sampled breast cancer datasets. We obtained higher mapping scores in metrics linked to the cell state continuum (cell, niche, territory). For each best-mapped patient pair, we also calculated the correspondence between their respective clinical metrics. 80% of samples share at least 3 of the 5 clinical metrics observed (PAM−50, ERRBB_2 positive, Grade, ER Status, and deathBreast). **b** Across the 5 clinical metrics, we observed that ERBB2_pos is the metric that is the most accurately recovered (89%) while PAM50 (cancer subtype) is the most poorly predicted (49%). **c** Clustering using ER-negative breast cancer samples. The

distinction of these samples is mainly driven by cell label and niche composition, which contrasts with the continuous cell state metrics. Clusters 9 and 10 – for instance – show little to no difference in cell, niche, and territory similarity in contrast to the stark difference in cell label and niche composition scores. **d** We highlight the clinical metrics associated with each cluster. We use the term Boolean to define True (1) or False (0). Cluster 9 is predominantly characterized by the HER2 cancer subtype, and cluster 10 is characterized by the Basal cancer subtype. Source data are provided as a Source Data file.

ERBB2-positive patients, even though we used only 37 protein markers (Fig. 6). These results suggest that patient status can be at least partially encoded in a cancer's multi-scale organization.

A key aspect to consider when using our approach is which cost matrix to utilize to obtain optimal cell mapping. In Supplementary Fig. S12, we demonstrate how each cost matrix may contribute to discriminating between cells and participate in the total cost. Discrete and categorical cost matrices (cell composition and cell type labels) tend to have a stronger effect on optimal mapping as seen in Figs. 2 and 3, and Supplementary Fig. S12. The assumption is that cell type labels (and by extension, niche composition) accurately represent prior biological knowledge. Our algorithm uses this prior biological knowledge to its advantage without requiring explicit weighting of each cost matrix. Across all our benchmarking, we observed that using all available information (cell, niche, territory similarities, niche composition, and cell type) provides the best combination for recovering center cell identity and cellular context. With that said, if cell type labels are not available or are not deemed appropriate (e.g., different nomenclature), using cell and niche similarity can still provide highly accurate mapping of cells across samples. We recommend including territory information since cells can express global context markers as we previously demonstrated[24]. Nevertheless, it is crucial to tailor any analysis to the question at hand.

Vesalius provides a mapping strategy with exceptional flexibility and interpretability, opening the door to tailored analysis utilizing all available information, including marker gene activity or cell fate information. We further investigated running time and memory usage over the number of cells (Supplementary Fig. S10 a/b). The running time and the memory usage are also dependent on the configuration used for the cost metric (Supplementary Fig. S10 c/d). Vesalius aims at understanding spatial biology across contexts and scales and provides a unique toolkit to investigate spatial samples as exemplified in our analysis related to cell states during brain regeneration, development, and patient classification. Our results demonstrate how crucial it is to consider multiple scales and contexts to understand the spatial relationship between cells.

## Methods
### Mapping cells across samples
To map cells across samples, Vesalius employs the Jonker-Volgenant algorithm $(O(n^3))$[35], a variant of the LAP. The algorithm is to map "workers" to "task" while minimizing the overall cost. To allow many-to-many matching of cells, we implemented the solver in a divide-and-conquer framework, which also provides a speedup for large batches through parallelization.

Given a requested batch size $B$, we randomly sample cells from the reference set $R$ and the query set $Q$. We select a subset $R_B \subset R$ and $Q_B \subset Q$ such that $|R_B| = |Q_B| = B$. Once a cell is selected, it is removed from the sampling pool to ensure all cells are eventually selected. If the number of cells in one dataset is smaller than in the other, padding is applied. Without loss of generality, assume $|R| < |Q|$. In this case, we create a duplicated set $R'$ and concatenate it with $R$, forming $R^* = R \cup R'$, where $|R^*| \geq B$ or matches $|Q|$, whichever is smaller. The Linear Assignment Problem (LAP) solver is then applied to each batch to optimize the mapping between the sampled reference and query cells. Given a cost matrix $C$ for a batch, the optimal assignment $M^*$ is found by minimizing

the total cost:

$$M^* = \arg\min \sum_{(i,j) \in M} C_{ij} \tag{1}$$

If a one-to-one matching is required, duplicate matches can be filtered by retaining only the pairs with the lowest overall cost.

Since the mapping depends on the cells selected during sampling, the batching and solving steps can be applied across multiple epochs. At each epoch $t$, new random batches $R_B^{(t)} \subset R$ and $Q_B^{(t)} \subset Q$ are selected and mapped using the LAP solver. Let $M^{(t)}$ represent the set of matched cell pairs at epoch $t$. The newly matched cell pairs are compared to those from previous epochs, and a pair $(i, j)$ is retained if it results in a lower matching cost:

$$C_{ij}^{(t)} < C_{ij}^{(t-1)} \tag{2}$$

where $C_{ij}^{(t)}$ is the cost associated with matching cell $i$ from $R$ to cell $j$ from $Q$ at epoch $t$.

The final output is a many-to-many mapping of cells across samples, along with the mapping cost, epoch information, and mapping scores for each cell pair.

### Computing cost
To solve the LAP, we compute a total cost matrix constructed from comparing cells through a variety of biological features. The default features used by Vesalius are the following:

*Cell similarity* (*f*): compares the gene expression of cells between each sample.

*Niche similarity* (*n*): compares the gene expression of cell niches between each sample.

Optionally:

*Territory similarity* (*t*): compares the gene expression of territories between each sample.

*Cell label* (*y*): checks whether both cells share the same cell type label.

*Composition similarity* (*c*): compares the cell type composition of cell niches between each sample.

*Custom similarity*: compares cells between samples using user-defined similarities.

To generalize the formula for an arbitrary number of similarity matrices, we define a set $n$ of similarity matrices $S^{(k)}$ for $k = 1, \ldots, n$. The final cost matrix $C$ is then computed as the sum of the reciprocals of all provided similarity matrices:

$$C_{ij} = \sum_{k=1}^{n} (1 - S_{ij}^{(k)}) \tag{3}$$

where $S_{ij}^{(k)}$ represents the similarity score between query cell $i$ and reference cell $j$ in the $k$-th similarity matrix. This formulation allows the incorporation of any number of similarity metrics. Importantly, cost matrices are not weighted. Cost matrices using discrete/categorical inputs (niche composition and cell type labels) leverage prior

biological knowledge and will yield a stronger discriminative ability without needing to be explicitly weighted.

In the case of continuous features (cell, niche, and territory), we first extract a gene expression signal. The signal represents a gene expression profile (normalized or raw) of all genes, highly variable genes, or a custom gene set. Alternatively, embedding values (PCA, UMAP, LSI, NMF) can also be used to define the gene expression signal. By default, and across all analyses presented here, we used the highly variable genes. Since genes might not perfectly overlap between datasets, we select the genes that lie at the intersection of gene expression signals for each dataset.

We calculate a Pearson's correlation coefficient between expression signals of each potential cell pair:

$$r = \frac{\sum (x_i - \bar{x})(y_i - \bar{y})}{\sqrt{\sum (x_i - \bar{x})^2 \sum (y_i - \bar{y})^2}} \tag{4}$$

where $x_i$ and $y_i$ are the expression of genes $i$ in sample $x$ and $y$, respectively. $\bar{x}$ and $\bar{y}$ are the mean expression of cell $x$ and cell $y$.

For categorical features such as niche composition, we compute a frequency-aware Jaccard index:

$$J(A, B) = \frac{|A \cap B|}{|A \cup B|} \tag{5}$$

We assign a unique label to duplicated cell type labels. If cell type A is present more than once, the first instance the label will be A, the second instance A.1, and so on.

Similarity matrices, which are computed using Pearson's correlation coefficient or a Jaccard Index, can be filtered by a user-defined threshold prior to mapping. For all cells in the query, we check if it has at least one match higher than the threshold.

We define an indicator function for the label agreement between a reference cell $j$ and a query cell $i$. Let $L_i$ and $L_j$ be the labels of the query cell $i$ and reference cell $j$, respectively. Then, we define:

$$I_{ij} = \begin{cases} 1, & if\, L_i = L_j \\ 0, & if\, L_i \neq L_j \end{cases} \tag{6}$$

This function assigns a value of 1 if the labels match and 0 otherwise. A query cell $i$ that satisfies the following condition:

$$\sum_i I_{ij} = 0 \tag{7}$$

Is assumed not to have any shared cell type labels with the reference data and can be removed upon user request prior to mapping.

## Extracting niches

For each cell, we define a niche as the spatially neighboring cells using the following methods:

*KNN*: K-Nearest Neighbors

*Graph*: Using Voronoi Tessellation, we define a neighborhood graph and select niches by considering the graph depth from a center cell, e.g., direct neighbors will have a graph depth of 1.

*Radius*: All cells within a certain radius of a center cell are defined as this cell's niche.

In most cases, we recommend using *Graph* since this method best captures the biological realities of cellular organization. We urge any user to carefully consider their question and tailor their analysis (and parameter selection) to the question at hand.

In the context of niche similarity, let $X_c$ be the gene expression profile of cell $c$, and $N(c)$ be the set of cells in the neighborhood of cell $c$.

Mean gene expression profile of the neighborhood for cell $c$:

$$\bar{X}_{N(c)} = \frac{1}{|N(c)|} \sum_{c' \in N(c)} X_{c'} \tag{8}$$

where $|N(c)|$ is the number of cells in the neighborhood of $c$, and $X_c'$ is the gene expression profile of cell $c'$.

Gene expression profile of cell $c$ becomes the average expression of its neighborhood:

$$X_c = \bar{X}_{N(c)} \tag{9}$$

## Extracting territories

Originally, Vesalius was designed to detect tissue territories from spatial transcriptomics data using image processing techniques. We have since improved the territory detection abilities. In addition to a wider range of dimensionality reduction techniques (PCA, UMAP, Latent Semantic Indexing (LSI), Non-Negative Matrix Factorization (NMF), custom) and normalization methods (log-normalization, SCtransform, Term-Frequency Inverse Document Frequency (TF-IDF), custom), we have adapted the image processing steps of Vesalius to function on image tensors instead of RGB formatted images. This upgrade allows us to apply the image processing steps to all latent space dimensions simultaneously and, in turn, improve our ability to recover tissue territories.

In summary, spatial transcriptomics data is converted into a grayscale image stack. We normalized counts and extracted highly variable features. With these features, we compute a reduced dimension latent space for each cell. In parallel, we expand punctual coordinates using Voronoi tessellation and rasterize the resulting tiles to produce a grayscale image for each latent space dimension. This image stack can be processed using a variety of image processing techniques such as histogram equalization, image smoothing, and image segmentation. Color segments can be further divided to distinguish spatially distinct color segments. The details of this process are discussed in our previous publication[24].

The territories isolated by Vesalius are used to compare the expression of cells across larger-scale spatial domains. We compare the average expression of the territory in which a cell finds itself across samples.

Let $T$ be a territory defined by Vesalius. Let $X_c$ be the gene expression profile of cell $c \in T$, and $T(c)$ be the set of cells in Territory $T$.

Mean gene expression profile of the neighborhood for cell $c$:

$$\bar{X}_{N(c)} = \frac{1}{|T(c)|} \sum_{c' \in T(c)} X_{c'} \tag{10}$$

where $|T(c)|$ is the number of cells territory $T$, and $X_c'$ is the gene expression profile of cell $c'$.

Gene expression profile of cell $c$ becomes the average expression of its neighborhood:

$$X_c = \bar{X}_{T(c)} \tag{11}$$

## Spatial simulations

To benchmark our mapping approach, we elected to use simulated data since they provide a strong ground truth against which we can compare performance. We used 5 regimes: *circle, layer, dropped, random one cell*, and *random two cell*. For each regime, we generate 12 samples, each containing 5000 cells and 2000 genes. We used the

splatter[42] package to generate synthetic single-cell-like count data for all samples. All cells for each cell type across each sample are taken from the same synthetic distribution. Reproducible generation of synthetic samples is available in a dedicated R package called oneiric (see "Code availability" section). The vignette of the package shows how to generate different samples as well as the data used for this analysis. The code is also available in the GitHub repository dedicated to the analysis presented here (see "Code availability" section).

For the *circle* regime, we created 5 circular territories of random size placed at random within a background territory. We allowed overlaps between territories but ensured that all territories were present across 12 samples. Each territory contains 2 cell types, and there is a probability $p = 0.5$ of differential gene expression between cell types.

For the *layered* regime, we created 1 circular territory of random size placed at random within a background territory. The circular territory is divided into 5 layers. The background and each layer will contain 2 cell types. There is a probability $p = 0.5$ of differential gene expression between the cells in the background and the circular territory. There is a probability $p = 0.05$ of differential gene expression between cells in each layer.

For the *dropped* regime, we created 2 circular territories of random size placed at random within a background territory. Each circular territory can be divided into 1–3 layers (the number of layers selected randomly). The background and each layer can contain 1 or 2 cell types. The selection of the number of layers and the number of cells is done on a sample basis, meaning that cells and layers will not necessarily be shared across samples. Cells from the same layers or cell type within a layer will always be sampled from the same statistical distribution. There is a probability $p = 0.5$ of differential gene expression between the cells in the background and the circular territories. There is a probability $p = 0.05$ of differential gene expression between cells in each layer.

For a *random one cell* and a *random two cell*, we created a background territory where a random number of territories (1–3), of random shape (circular, rod, or chaos map), with a random number of layers (0–4) will be placed. *Random one cell* will only contain at most one cell. *Random two cell* will contain at most 2 cells (number of cells randomly selected). There is a probability $p = 0.5$ of differential gene expression between the cells in the background and territories. There is a probability $p = 0.05$ of differential gene expression between cells in each layer.

For each synthetic regime, we exported the cell labels of each cell as well as the neighborhood interactions. Simply put, we select the K-nearest neighbors ($k = 6$) and concatenate the labels of the neighborhood into an "interaction" label (using @ as separator for discrimination later). Adding interactions is part of the oneiric package with the *add_interactions* function. Data generation code is available on the GitHub dedicated to the analysis presented in this manuscript (see "Code availability" section).

In addition to these datasets, we also generated 3 samples of the circle regime with varying numbers of cells (100, 500, 1000, 2500, 5000, 10,000, 20,000, 50,000, 100,000, 200,000). These datasets were used for computational performance benchmarking (see below).

### Biological data preparation for benchmarking

For format consistency, we prepared and processed each biological dataset used for benchmarking, namely, seqFISH mouse embryo, Slide-seq V2 mouse hippocampus, and Stereo-seq mouse embryo (see "Data availability" section).

We split datasets into slices when necessary. seqFISH is available as a single R object, which we split into 3 mouse embryo sections. This step was not necessary for Slide-seq V2 and Stereo-seq since they are provided as separate files. For all datasets, we rescaled spatial coordinates to ensure that the minimum x and y coordinates would be 1. Next, using oneiric, we added neighborhood interactions labels with

KNN ($k = 10$). All datasets are exported in the same format, consisting of a coordinate data frame with 6 columns (barcodes, x, y, sample, cell_labels, interactions) and a gene-by-cell count matrix.

For Slide-seq datasets, we first filtered cells that contained less than 200 total counts. We also filtered genes with less than 100 total counts across all cells. Then, we annotated datasets using RCTD[37] with the same mouse brain scRNA-seq reference data[38] in order to quantitatively assess cell and spatial context mapping abilities. For Stereo-seq, we downsampled to 50,000 cells to reduce run time.

Data preparation code is available on the GitHub dedicated to the analysis presented in this manuscript (see "Code availability" section).

### Data preparation of cross-technology and cross-resolution mapping

To compare cells across technologies, we processed seqFISH[34] and Stereo-seq data[9]. seqFISH data was filtered by removing low-quality cells as described by the authors. Next, we pre-processed both datasets using Vesalius by log-normalizing the counts and producing a reduced dimension latent space using PCA (PCs = 30). Our cross-technology comparison only used cell similarity and niche similarity; as such, no further processing was required.

Matching cells across resolutions required extra steps for data preparation. First, we downsampled the VisiumHD mouse brain data to retain 200,000 cells. We then used RCTD in doublet mode to deconvolve and assign cell type labels. As a reference, we used the labeled scRNA data provided by ref. 38. We remove all cells that contain the "reject" tag. For all remaining cells, we annotated cells with one or two labels depending on how RCTD defined them (singlet or doublet). After this filtering, the VisiumHD data contained 84,287 cells. We also used RCTD to deconvolve the Visium mouse brain data with the same reference dataset[38]. In this instance, we only retained cell type labels that were considered "high confidence".

### Benchmarking mapping performance

We contrasted our mapping approach to single-cell mapping tools, spatial alignment tools, and data integration tools. More specifically, we compared Vesalius to CytoSpace[35], Tangram[35], PASTE[27], GPSA[28], SLAT[33], and Scanorama[32]. It is of note that CytoSpace requires perfect matching of cell type labels between reference and query and only finds a proposed mapping of cells between the same cell type labels. Importantly, if the cell labels are not shared, CytoSpace will not run to completion. To demonstrate how this strategy differs from our context-specific approach to cell mapping, we provided CytoSpace with true cell labels as well as a single-cell label for all cells (noted with the _noLab label).

For Vesalius, we ran mapping with 14 cost matrix combinations:
- feature
- niche
- territory
- composition
- feature-niche
- feature-niche-cell_type
- feature-composition
- feature-territory
- niche-territory
- niche-composition
- feature-niche-territory,
- feature-niche-composition
- feature-niche-composition-territory
- feature-niche-composition-territory-cell_type

Where features represent cell similarity, niche represents niche similarity, territory represents territory similarity, composition represents cell type composition of niche, and cell type represents the cell label score.

To quantitatively assess the performance of cell mapping between samples, we computed an Adjusted Rand Index[36] (ARI) between the cell labels of the query and the reference. We also computed a Jaccard Index between the cell labels of the neighborhood of mapped cells to ascertain if we were capturing the spatial and cellular context of each mapped cell. Since PASTE and GPSA do not match cells but rather modify the coordinates of each cell, we computed the nearest cell neighbor between the reference and query and compared their cell type labels. While this approach disadvantages PASTE and GPSA, our objective is to demonstrate that context mapping is a fundamentally different task from spatial alignment. Scanorama is a tool designed to integrate single-cell datasets; it only provides an integrated latent space, but not best-matching cells. As such, we computed the nearest neighbors between the reference data and query data in the integrated latent space to determine best-matching cell.

For Vesalius (==2.0.0), we ran our mapping using 1000 cells (for synthetic data) or 5000 cells (for real data) across 25 epochs. We set a cell filtering threshold of 0.9 and allowed cell type label filtering if they were provided. We used the log-normalized counts to define the gene expression signal. Niches were defined using KNN with $k = 6$ for synthetic data and $k = 10$ for real data. In both cases, we select the top 2000 variable genes for analysis.

For CytoSpace (==1.0.6a0), we reformatted synthetic data and real data to fit the CytoSpace data requirement. We used 5000 cells as a batch size since CytoSpace down-samples based on cell type labels. We ran CytoSpace using their single-cell mode (does not require the deconvolution step to be run).

For Tangram (tangram-sc ==1.0.4), we loaded data as AnnData objects. We log-normalize the data and extract the top 2000 variable genes. We use these variable genes as seed genes for Tangram mapping. We use "rna_count_based" as the density prior and ran Tangram for 10,000 iterations.

For PASTE (paste-bio==1.4.0), we loaded data as AnnData objects. Data objects do not need to be normalized prior to alignment. We ran PASTE for 10,000 iterations.

For GPSA (==0.8), we loaded data as AnnData objects. We ran GPSA for 50 iterations as recommended by the authors. The learning rate for the Adam optimizer was set at 0.01 as recommended by the authors.

For SLAT (scslat==0.2.1), we loaded data as AnnData objects. SLAT does not require data to be normalized prior to mapping. We selected neighborhoods using KNN ($k = 6$ for synthetic data and $k = 10$ for real data). We ran SLAT with 5 Graph Convolutional layers across 25 epochs. Parameters were modified in accordance with developer recommendations.

For Scanorama (==1.7.4), we loaded data as AnnData objects and ran Scanorama with default parameters.

For all benchmarking, all code, parameters, and dependency lists used in this analysis are available in the dedicated GitHub.

## Computational performance benchmarking

We included benchmarking for computational performance by varying the number of cells used during mapping (100, 500, 1000, 2500, 5000, 10,000, 20,000, 50,000, 100,000, 200,000) in a circle synthetic regime.

For this task, we aimed at making the performance as fair as possible; as such, we assumed that a user would run a full analysis pipeline, including data loading, preprocessing, and mapping on a single CPU. We made this assumption since all tools have different input requirements, different downstream tasks, and different optimization options. Vesalius provides the widest variety of potential downstream tasks and, as such, has the most requirements in terms of preprocessing.

We also aimed to avoid any variation in language-specific methods of measuring run time and memory usage. We measured run time and memory usage (Resident Set Size – peak RAM usage) for each tool using custom bash scripts (Available on the dedicated GitHub page) that measured these metrics across 3 submitted jobs. We set hard computational limits: a maximum of 480 GB of RAM and a maximum run time of 12 h. If the job exceeded these limits, the run was cut short.

We compared run time and memory usage across cost matrix combinations for 5000 and 10,000 cells. We used the same 14 matrix combinations described above and used the same performance measuring strategy as described above.

## Cross-technology and cross-resolution mapping

To map Stereo-seq mouse embryo to seqFISH and vice versa, we mapped cells in batches of 10,000 cells across 10 epochs. Since our approach selects the intersections of genes available between both datasets, we use all genes (135 genes in the seqFISH datasets).

For cross-resolution mapping, we constructed the cost matrix using niche and territory. We defined territories by processing each dataset through the Vesalius pipeline, which includes color histogram equalization, image smoothing, and image segmentation. The parameter details are available in the analysis scripts deposited in our dedicated GitHub page (see "Code availability" section). We defined niches using a radius from the center cell, where the radius is defined as half of the scaled distance between Visium Spots. Cell matching was achieved across 20 epochs with a batch size equal to the number of Visium spots ($n = 2310$). We used log-normalized variable features as the expression signal to compare. Once the matching was completed, we ran the mapped dataset through the Vesalius pipeline to retrieve spatial domains. We added a jitter to cell coordinates to enable each mapped cell to have a distinct spatial coordinate. In contrast, no jitter will merge the mapped cells into Visium-like spots. To compare the cell types mapped across resolution, we first scaled the cell type frequencies in the deconvoluted Visium Spots to assume 100 total cells in each spot. We applied the same frequency scaling to the cell type mapped from VisiumHD. We employed this strategy since RCTD returns an estimated cell type proportion and not a number of cells, and the number of cells can vary drastically between spots. Using these scaled cell type frequencies, we constructed a contingency matrix upon which we applied a Chi-squared test to determine if the cell type frequencies were statistically distinct. In addition, we computed a Jaccard Index between cell type labels in the Visium spots and mapped VisiumHD cells to determine how many cell types were being missed despite having potentially low occurrence frequencies.

## Count integration

Once cells have been matched across samples, we can integrate the counts and develop an integrated tissue analysis framework. Count integration was only required during the analysis related to Stereo-seq Axolotl brain regeneration and Stereo-seq Mouse embryo across developmental time. To integrate counts, we used the Seurat implementation of Canonical Correlation Analysis[43] (CCA), which returns scaled and integrated counts and a common latent space between samples. Vesalius can utilize all methods deployed by Seurat (e.g., scVI or Harmony). The integrated latent space is directly used to generate Vesalius images upon which our pipeline can be applied. To avoid duplicated coordinates, we added a jitter to the duplicated coordinates only. The resulting territories are the territories emerging from this joint latent space, which can be used for spatially resolved differentially expressed gene expression analysis.

For inter-sample differential gene expression analysis, Vesalius can compare the cells from the integrated count matrix using either territories or cells as a grouping criterion. By default, we used the Wilcoxon ranked sum test with the p-value threshold set at 0.05 after correcting for multiple testing (FDR).

## Cell mapping clusters

We developed a cell clustering method that clusters query cells based on which reference cells they tend to co-map to. Our default clustering approach is based on hierarchical clustering with community-based clustering approaches (Leiden and Louvain) also available. First, we define the metric matrix that will be used for clustering. This approach follows the same pair-wise summation of similarity scores used during cost matrix creation described above. By default, we use cell similarity only. We define co-mapping of cells using the following steps:

Order of preferential mapping

For each cell $c$, let $M(c) = \{m_1, m_2, \ldots, m_k\}$ be the set of mapping costs where $m_i$ represents the cost of mapping to reference cell $i$. The preferential mapping order is given by sorting $M(c)$ in descending order:

$$M^*(c) = \text{sort}(M(c)) \tag{12}$$

The numerical values are converted into categorical strings.
Selection of top $n$ matches
We select the top $n$ matches (default $n = 30$):

$$S(c) = \{m_1^*, m_2^*, \ldots, m_n^*\} \tag{13}$$

Where $S(c)$ represents the categorical vector of the top $n$ mappings for cell c.

Jaccard Index Computation

Given two query cells $c_1$ and $c_2$, their Jaccard Index is computed as:

$$J(c_1, c_2) = \frac{|S(c_1) \cup S(C_2)|}{S(c_1) \cap S(C_2)} \tag{14}$$

A higher $J(c_1, c_2)$ indicates a greater likelihood that two query cells map to the same set of reference cells.

Distance Matrix for Hierarchical Clustering

The reciprocal of the Jaccard Index is used as the distance matrix for hierarchical clustering or community clustering algorithms.

$$d(c_1, c_2) = 1 - J(c_1, c_2) \tag{15}$$

For clarity, we used a fixed number of clusters with $k = 5$ in the context of Stereo-seq mouse embryo clustering.

## Mapping across spatially heterogeneous prostate cancer samples

We mapped cells from 2 prostate cancer samples using Slide-seq V2 technology[44]. In this instance, we downloaded the count matrices, spatial barcodes, and cell type annotations provided by the authors. We processed 2 tumor samples (Tumor01 and Tumor02) using the Vesalius pipeline. In brief, we normalized gene counts, selected variable features ($n = 2000$), and reduced dimensionality via PCA (30 PCs). Next, we smoothed the Vesalius image stacks and equalized the color histogram before segmenting the images and isolating territories. We mapped each sample to each other using cell, niche (KNN with $k = 15$), territory similarities, and cell type labels. The mapping procedure was carried out across 25 epochs with a batch size set to 5000, consistent with our previous analysis. The details of the analysis are available on the dedicated GitHub.

## Spatial proteomics in situ mass cytometry sample preparation and processing

To demonstrate the utility of Vesalius, we mapped spatial in situ mass cytometry (IMC) data taken from human breast cancer patient samples[41]. Out of the 709 samples, we filtered the samples to ensure that they contained at least 1000 cells. We also filtered out samples that contained NAs in the clinical metrics provided by the authors (ER

Status, ERBB2_pos, grade, deathBreast, PAM50, and YearsToStatus). From the remaining samples, we randomly sampled 50 ER-positive and 50 ER-negative samples. Next, we extracted the expression values for the 37 markers measured, which we formatted into a count matrix for Vesalius. We processed every sample using the Vesalius pipeline, including log-normalization, dimensionality reduction (PCA) and image processing, and territory isolation. The details of the parameters used are contained in our analysis scripts deposited in a dedicated GitHub page (see "Code availability" section).

We mapped cells between samples using a cost matrix built from cell similarity, niche similarity, territory similarity, cell label, and niche composition. We optimized the matching pairs across 10 epochs with a batch size equal to the smallest dataset minus one (or 1000 - 1 cells if both datasets were larger than 1000 cells). The neighborhood was defined through our graph method with a depth of 2. For each mapping event, we extracted the mean cost and mean mapping scores.

For each sample, we selected its best-matching pair (excluding self-mapping) by taking the pair with the lowest overall cost. From this best-matching pair, we computed the overlap between the clinical metrics to check whether our mapping strategy could recover patient-level information. More specifically, we checked if we obtained the same label between the reference and query. In the case of Grade and PAM50, there are multiple labels: "1", "2", "3" for Grade and "Basal", "HER2", "Luminal A", "Luminal B", "Normal-like" for PAM50. The delta *YearsToDeath* was calculated by taking the absolute value of the difference in years between query and reference.

$$\Delta = |Q - R| \tag{16}$$

## Sample mapping clusters

To cluster samples (IMC), we first mapped samples to each other, extracted their average cost, and averaged mapping scores across all cells in those samples. Since we are mapping samples to each other, we can directly use the mapping cost matrix as input to the hierarchical clustering algorithm. In the case of IMC, we used height to define the number of cluster cutoff ($h = 0.5$).

## Reporting summary

Further information on research design is available in the Nature Portfolio Reporting Summary linked to this article.

## Data availability

All datasets used in this study are publicly available datasets. We provide the URLs below. Simulated datasets can be generated using the dedicated R package. Source Data are provided with this paper. *Simulated data*: https://github.com/WonLab-CS/oneiric *Slide-seq V2*: https://singlecell.broadinstitute.org/single_cell/study/SCP815/sensitive-spatial-genome-wide-expression-profiling-at-cellular-resolution#study-summary *seqFISH*: https://content.cruk.cam.ac.uk/jmlab/SpatialMouseAtlas2020/ *MOSTA*: https://db.cngb.org/stomics/mosta/stereo.seq/ *Visium*: https://cf.10xgenomics.com/samples/spatial-exp/2.0.0/CytAssist_FFPE_Mouse_Brain_Rep1/CytAssist_FFPE_Mouse_Brain_Rep1_web_summary.html *VisiumHD*: https://www.10xgenomics.com/datasets/visium-hd-cytassist-gene-expression-libraries-of-mouse-brain-he scRNA: Molecular Diversity and Specializations among the Cells of the Adult Mouse Brain[38]. *ARTISTA*: https://db.cngb.org/stomics/artista/ Prostate Cancer: Dissecting the immune suppressive human prostate tumor microenvironment via integrated single-cell and spatial transcriptomic analyses[44] *IMC*: Breast tumor microenvironment structures are associated with genomic features and clinical outcome[41] Source data are provided with this paper.

## Code availability

The Vesalius package used in this study is publicly available and has been deposited in GitHub at https://github.com/WonLab-CS/Vesalius under GPL-3.0. The Oneiric package used to generate the synthetic data is publicly available and has been deposited in GitHub at https://github.com/WonLab-CS/oneiric under GPL-3.0. The code used to perform analysis and generate results is publicly available and has been deposited in GitHub at https://github.com/WonLab-CS/Vesalius_analysis under GPL-3.0. The specific version of the code (packages and analysis) associated with this publication is archived in Zenodo and is accessible via: https://doi.org/10.5281/zenodo.15733817[45]

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

## Acknowledgements

The authors gratefully acknowledge the support of Cedars-Sinai Medical Center through institutional funding. This study was also supported by VCU Massey Cancer Center and School of Medicine start-up funds, VCU, cMEDA Seed Award, the Adenoid Cystic Carcinoma Research Foundation, the Swiss Cancer League, Seeds of Science, SNSF, Fundação para a Ciência e a Tecnologia, Fundamental Mandates (Stichting tegen Kanker—Fondation contre le Cancer), and CSGS, 3P30CA016059–42S1, NCI, NIH to R.G.; VCU Massey Cancer Center and School of Medicine start-up funds and Tina's Wish Foundation Award and CSGS, 3P30CA016059–42S4, NCI, NIH to E. Madan.; R.G and P.B.F acknowledge the support for this project and program from the VCU Institute of Molecular Medicine (VIMM).; support from NIH/NCI R01 CA280194 (PBF).

## Author contributions

P.C.N.M. and K.J.W. conceptualized the present study. P.C.N.M., W.W., and H.H. developed the methodology and software. P.C.N.M., W.W., and H.K. performed analysis, including benchmarks, data annotation, and data visualization. P.C.N.M., K.J.W., and R.G. wrote the draft of the manuscript. K.J.W. and R.G. supervised the project. Reviews and editing were performed by P.C.N.M., W.W., H.K., H.H., P.B.F., A.P.S., R.W., E.M., R.G., and K.J.W.

## Competing interests

The authors declare no competing interests.
