## [Transparent Peer Review file · Nature Communications]

Multi-scale and multi-context interpretable mapping of cell states across heterogenous spatial samples

Corresponding Author: Dr Rajan Gogna

Version 0:

Reviewer comments:

Reviewer #1

(Remarks to the Author)

Spatial data across tissue contexts may be difficult to compare due to a lack of discernible structural similarity. Indeed, particularly in human clinical tissue settings, it is difficult to expect tissues to share similar structures, particularly at the scale that is evaluated. In this manuscript, while Martin et al acknowledge this challenge, they present a solution called Vesalius that relies on aligning samples based on minimizing a cost function that involves quantifying multiscale and multi-context compositional variation that capture cell similarity, niche similarity, and spatial territory similarity. We were generally confused how the proposed solution addresses the presented problem because it ultimately still relies on structural similarities. So while the presented problem is an important one, we fail to see how the developed approach provides a solution. We hope these comments will be useful to the authors moving forward:

1. It was not clear if the algorithm included any regularization constraints to ensure spatial continuity of the alignment. If not, it sounds somewhat like other single-cell batch correction algorithms like scanorama (<https://www.nature.com/articles/s41587-019-0113-3>) focused on aligning single cell manifolds and should be compared appropriately
2. Likewise, the authors mainly mention rigid alignment approaches like PASTE. How does Vesalius compare to diffeomorphic (non-rigid) alignment techniques like Spateo (<https://www.biorxiv.org/content/10.1101/2022.12.07.519417v1>), STalign (<https://www.nature.com/articles/s41467-023-43915-7>), cst(<https://www.nature.com/articles/s41592-024-02410-7>), etc?
3. In general, the accuracy of the alignment on real spatial transcriptomics data was unclear. The authors need to include more quantitative evaluations.
4. The Vesalius algorithm appears to scale as $O(n^3)$ and $O(n^4)$, which is highly inefficiency and would take a very long time to run, especially given newer higher resolution and generally larger datasets. How long does it take to run as a function of the number of cells and labels, niches, territories, etc, evaluated?
5. In general, the authors need to address the presented problem by comparing multi-scale and multi-context trends of cell states across heterogenous spatial samples. There are many such datasets publicly available even. For example, consider different cancer samples from different patients that are completely structurally different.

(Remarks on code availability)

- The website link is broken <https://patrickcnmartin.github.io/Vesalius/> giving a 404 error

- The vignettes while existent are not compiled, making it difficult for us to evaluate whether the code is functional

Reviewer #2

(Remarks to the Author)

This paper proposes a computational tool developed to align spatial data, from the same technology, from adjacent tissue sections, resulting in effective cell, tissue and niche mapping strategies. Additionally, the tool demonstrated relatively effective mapping between different spatial technology assays (Stereo-seq and seqFISH) and different assay resolutions

(Visium and Visium HD).

The rapid rise of spatial transcriptomic technologies has resulted in large amounts of complex data generated with few analysis methods tailored to the type of data produced. Creating novel analysis tools such as this cell mapping method, are essential if the spatial transcriptomics field is to last. In particular, methods focused on alignment or integration are currently rare, though the need for this is growing, making the significance of this paper important. This paper claims to be able to map cells across adjacent tissue sections, which it succeeds in doing so, recovering tissue structures within brain and embryo samples. It may be of interest to test this tool on less structurally sound datasets such as cancer samples. However the current trends when developing any spatial transcriptomic associated research is to test on brain samples, the samples selected are well within the standards of the field.

A few suggestions would be;

Include some statistics along with figure 2a boxplots when comparing the different method performances.

Is there some type of mapping score in figure 3.e that could be done to test between the different technologies?

Colour legend is missing from a few of the figures (figure 3.c and 3.f).

When looking at Visium and Visium HD cell type spot overlap (figure 3.f), a percentage of spots with a high Jaccard index would help clarify and strengthen the statement 'most spots contain the same cell types between technologies' made. E.g. 70% of spots have an index of 0.7 and above.

In figure 4 (a and d) it would be beneficial to have the labels for 'reference', 'mapped' etc.

Figure 5c is not referred to in the text, even though the results are discussed which initially led to some confusion.

Overall, this is a highly beneficial paper that has taken the task of generating a possible tool to aid in tissue alignment, something that is highly needed in the field of spatial transcriptomics analysis.

(Remarks on code availability)

Reviewer #3

(Remarks to the Author)

The authors have developed a tool previously that is able to identify spatial structures in spatial molecular data. The manuscript here is an extension, in which the authors leverage the so-called territories, to integrate spatial molecular data with a query dataset. For this task, the authors implement a linear assignment strategy as part of the previous package. The authors suggest that this strategy outperforms competing strategies.

General comments

Personally, I find this work exciting and looking forward to its publication. The phrase "I think there is something here" keeps popping into my head. It just needs to be cleanly worked out and shown. I find it crucial to assess it more carefully against other approaches, such that it also becomes clear towards the reader and user, in which situations this approach is going to be superior, and in which cases others maybe preferable. Computing and memory requirements is a major concern with these approaches, but also proper benchmarking should be done to provide a more complete picture. I am actually willing to believe that a LAP solution may beat a badly trained NN, but formally I like to see the evidence (see more detailed comments).

Major

-) Most evaluation is qualitative. Only limited quantitative benchmarks. I like to see more quantitative evaluations. There are already benchmark papers out there (e.g. <https://doi.org/10.1038/s41592-022-01480>). I generally do not believe that a novel approach needs to beat all tools in all scenarios to be great pieces of new research (in case this was the motivation not to systematically assess it).

-) Figure 2 contains comparisons to existing approaches only on simulated data. While comparison in a controlled dataset makes sense I would have liked to see performance differences on the real datasets as well.

-) To me the question how territory/compositional effects contribute towards matching is unclear. The break down in extended figure S1a and S1b seems inconclusive since the "feature" cost already provides a good solution. S1c shows that cell-type is the driving cost. It would be great to get more in-depth on this aspect since the whole point of integrating this approach into Vesalius implies that the point is to leverage this aspect.

-) Figure 2b on PASTE and GPSA seems inconsistent of how I would consider the ARI to look like in 2a. Do you have an explanation here?

-) Figure 2. It would be great to see what the supposed optimal mapping for the simulated datasets should look like to get a better idea about the point the authors are trying to make.

-) Details of settings of competing tools is missing. This is particularly problematic, since I think it should be transparent and clear what the differences are. E.g. removing the cell-type information from CytoSpace is in principle ok to make a specific point, but I would object to suppress results with that feature. It is misleading. I rather think the authors should have an open discussion on this feature then (which is also implemented in their own approach).

-) A discussion on the compute requirements compared also to the other approaches would be warranted, since current datasets are increasing in complexity.

-) There are different types of tasks demonstrated here. In principle, Vesalius can handle them, but it does so by switching on/off scenarios that influence the cost function. While I appreciate this flexible platform, from an evaluation perspective one should appropriately benchmark with the according competing tools.

Suggestions and Minor comments

-) I did find the manuscript difficult to read and I think this is due to the structure. Please feel free to ignore this comment if the authors feel strongly against this, however from a paper reading perspective, organising it based on the tools features rather than by technology may make it easier to read and follow (.e.g using LAP with default features, using LAP with default feature and optional feature etc)
-) Formatting is inconsistent at some places (e.g.: Supplemental Figure S2 vs Supplemental Figure 3)
-) Some grammatical mistakes here and there. I suggest to run it through an according grammar checker of your choice.
-) Vesalius citation has mark-down elements visible
-) Wish for more mathematical formulation of the method section (e.g.: concrete cost function). One can piece it together based on the information provide, but it would help with clarity of more technical people.
-) Introduction generally points out existing tools, but does not put it in context of the approach of the authors.
-) Figure 5 axis labels not readable

Final thoughts:

I think the authors have implemented a solution that could make a great contribution towards ST integration. My intuition and the results provided makes me think that this could actually work well, but it is formally not shown sufficiently. For that reason, I recommend this manuscript to be rejected in its current version, but strongly suggest to reconsider it once the according benchmarking work has been done.

(Remarks on code availability)

Version 1:

Reviewer comments:

Reviewer #1

(Remarks to the Author)

We thank the authors for the well organized and thorough explanations addressing our comments. The revised introduction clarifies the utility of this approach compared to rigid and diffeomorphic alignment algorithms that assume structural coherence. The new runtime benchmark is also greatly appreciated and helps set expectations for reasonable runtimes despite the big(O) complexity. We will look forward to trying out this tool on our data.

Our only major remaining question is: in the revised text, we now better appreciate that Versalius is incorporating a local spatial context to create a joint embedding. Other approaches also seek to create a joint embedding but doesn't take such spatial context into consideration as well explained by the authors. However, what is modulates the size of this spatial context that Versalius takes into consideration? From the methods, it looks like either a KNN, Graph, or Radius distance is used. Do the authors have recommendations/guidance on how to determine an appropriate niche size?

Particularly since the benchmarking is later done assuming a $k=6$ neighborhood size, we're wondering how these hyper parameters interact. For example, if we choose a niche size <6 , then will benchmarks assuming a $k=6$ neighborhood size in the benchmark suggest poorer performance?

Likewise, hearkening back to the title of the paper which focuses on multi-scale integration, how does Versalius parameters prioritize local versus global niche structures? For example, there may be some samples that are compositionally similar on a fine scale but rather different on a coarser/global scale, particularly in the context of cancer. Or alternatively, there may be diseased tissues of structurally stereotypic organs like brains that are quite similar on a coarser/global scale to healthy tissues but locally very different. Is Versalius able to guide users to the appropriate scale of similarity/differences?

A general discussion of limitations is warranted.

I was asked to comment on responses to Reviewer #3's comments.

Regarding Comment 1-2: The authors now provide more thorough benchmarks with real ST data, addressing these previously raised concerns.

Regarding Comment 3: The reviewer's previous comment highlights a point of confusion as to how territory/compositional effects contribute to the cost function, particularly since cell-type composition appears to be driving this cost function. While the revision includes equations defining the cost matrix and quantifying niches, territories, etc, it remains unclear how these various quantifications are weighted.

While all cost matrix combinations are shown, the interpretation remains unclear. A discussion on how to best use f_{ncty} , f_{nct} , etc modes of Vesalius is needed.

Regarding Comment 4: The revised text now clarifies the distinction spatial alignment approaches versus Vesalius

Regarding Comment 5: The revised figure addresses this previous concern

Regarding Comment 6: The reviewer was looking for the default settings used for competing tools to ensure reproducibility. It would be appropriate to include version number of default parameters in methods.

Regarding Comment 7: The new runtime benchmarks address this previous concern

Regarding Comment 8: This relates to Comment 3 and remains to be thoroughly addressed

All other minor comments have been addressed

(Remarks on code availability)

Looks reasonably well documented though we have not tried to install or reproduce the results.

We confirm the website is now functional and vignettes are compiled.

Reviewer #2

(Remarks to the Author)

The additions made to the paper are appropriate, clarifying some concerns raised by reviewers. In particular, the addition of benchmarking has greatly clarified the benefits compared to other similar tools e.g. matching cells from non-adjacent issue, and inclusion of spatial context to the model. Of note, they have included prostate cancer samples to test the viability of Vesalius in highly heterogenous spatial datasets, successfully demonstrating the benefit this tool would provide for spatial cancer research, which often works with multiple sections of FFPE tissue and has difficulty cell typing across tissue. Overall, the paper provides a spatial specific tool that would be highly beneficial in the current spatial biology landscape.

(Remarks on code availability)

Version 2:

Reviewer comments:

Reviewer #1

(Remarks to the Author)

The authors have thoroughly addressed the previous comments.

(Remarks on code availability)

We thank the reviewers for their constructive reviews. Please find below our response. We provide a summary of the changes as well as point-by-point responses. For clarity, we provide our response to reviewer comments in a color-coded manner. Blue represents the reviewer's concern, in black our response (if any) and in red the changes we made to address the comments. We also include the text changes and the newly generated figures (when appropriate). The text changes within the manuscript itself are highlighted in red.

Summary of changes

To enhance the value of a manuscript as suggested by the reviewers, we first adjusted our simulation data to better reflect the role played by spatial context in mapping cells across structurally heterogeneous spatial samples. We quantitatively benchmark our algorithm in highly heterogeneous synthetic scenarios against state-of-the-art alternate methods (SLAT, PASTE, GPSA, Tangram, CytoSpace, and Scanorama). We extend our benchmarking across 3 different spatial biology technologies (seqFISH, Slide-seq-v2, Stereo-seq). We also demonstrate computational performance in synthetic data sets with increasing number of cells (100 to 200k). Finally, we demonstrate the use of Vesalius in highly heterogeneous prostate cancer spatial transcriptomic data sets. Overall, we have aimed at grouping similar reviewer comments under larger scale changes.

Point-by-point response

Reviewer 1

Spatial data across tissue contexts may be difficult to compare due to a lack of discernible structural similarity. Indeed, particularly in human clinical tissue settings, it is difficult to expect tissues to share similar structures, particularly at the scale that is evaluated. In this manuscript, while Martin et al acknowledge this challenge, they present a solution called Vesalius that relies on aligning samples based on minimizing a cost function that involves quantifying multiscale and multi-context compositional variation that capture cell similarity, niche similarity, and spatial territory similarity. We were generally confused how the proposed solution addresses the presented problem because it ultimately still relies on structural similarities. So while the presented problem is an important one, we fail to see how the developed approach provides a solution. We hope these comments will be useful to the authors moving forward:

1. It was not clear if the algorithm included any regularization constraints to ensure spatial continuity of the alignment. If not, it sounds somewhat like other single-cell batch correction algorithms like scanorama (<https://www.nature.com/articles/s41587-019-0113-3>) focused on aligning single cell manifolds and should be compared appropriately

Our approach matches cells from different spatial samples under the assumption that these samples are *not* from adjacent tissue sections. We do not assume that samples are spatially continuous. In fact, this assumption would be incorrect in highly spatially heterogeneous samples which is our main aim. The advantage of our assumption was clearly demonstrated by mapping of cells in heterogeneous samples in our synthetic data and real data examples (See Figure 2,

Supplementary Figure S1-S9, Figure 3, Figure 4, Figure 5, Supplementary Figure S15). While our approach is still able to match cells from adjacent tissue sections, our cell matching is based on the spatial continuity with the neighboring cells (but not the entire tissue): the transcriptional similarity (e.g. correlation between gene expression profiles) of cells and “their niches”. By including niches, we can map cells with spatial information even though tissue structures are not same. The flexibility of our approach also allows the addition of territory level information. In this regard, our approach is different from scRNAseq integration tools such as Scanorama which does not include spatial context. As the reviewer suggested we included Scanorama in our benchmark to demonstrate the usefulness of using spatial context for mapping spatial data,

We demonstrated the advantages of our mapping strategy using expanded synthetic data (Figure 2 and Figure S6-S9) where Vesalius performed as good as other state-of-the art approaches. More importantly, the advantage of Vesalius was better highlighted when using real data (Figure 3 and S11). Vesalius far outperformed other approaches in mapping cell types and cellular context of the nearby cells for SeqFISH, Slide- Seq and Stereo-Seq data.

Fig. 3a

Fig. 3b

Fig. 3d

Fig. 3e

Fig. S11a

Fig. S11b

Figure R1. Benchmarking using real data demonstrate the outstanding mapping performance of Vesalius. It shows the comparison using SeqFISH, Slide-Seq, and SteroSeq data. The use of ARI is biased for Cytospace as Cytospace is designed to map cells with the same cell types. Using JI for cellular context demonstrates that Vesalius is much better suited at capturing the spatial context than any other tool including CytoSpace with labels.

We also would like to stress that Vesalius has the ability to incorporate various available spatial information. For instance, we demonstrate the performance of Vesalius across 14 cost matrix combinations (the combination of cell similarity (feature ; f), niche (n), territory(t), and cell type composition (c) and cell type labels (y)). In addition, this approach lends itself to adding new and context specific cost matrices.

Changes in text

Introduction

“Along the same lines, methods designed to integrate single cell data such as Scanorama[1] can find the closest related cells in their integrated latent space. However, data mapping or integration without spatial and biological context can lose the information related to tissue or cellular microenvironment.”

Results

“Benchmarking mapping performance in synthetic spatial data

To demonstrate the effectiveness of our mapping strategy, we elected to use synthetic spatial data since this will provide a robust ground truth (See methods). Specifically, we aimed to demonstrate our ability to accurately map cells across samples along with their spatial context. First, we generated 5 synthetic spatial regimes (circle, layer, dropped, random one cell, random two cell) that mimic the complexity of biological scenarios (Supplementary Figures S1-S5). For each regime, we generated 12 samples where we aimed to map a query data set on to a reference data for a total of 132 mapping events (excluding 12 self-mapping events). Details of each regime are developed in the method section and each sample can be visualized in Supplementary Figures S1-S5.

Since Vesalius provides a flexible framework to construct the total mapping cost, we compared the performance of Vesalius across 14 different cost matrices in addition to two scRNA-to-spatial mapping tools (CytoSpace[2] and Tangram[2]), spatial alignment tools (PASTE[3], and GPSA[4]) and scRNA/Spatial data integration tool such Scanorama[1] and SLAT[5]. The cost matrix combinations are built using cell similarity (f = feature), niche similarity (n = niche), territory similarity (t = territory), niche cell type composition (c = composition), cell type labels (y = cell type labels) or a combination of these metrics.

To measure the ability of each tool to effectively map cells across samples, we computed an Adjusted Rand Index (ARI)[6] between mapped cell labels. In parallel, we measured each tool’s ability to capture spatial

context by adding an “interaction” label to our data sets (see Methods). This label concatenates the cell type labels of k nearest neighbors ($k = 6$) of a center cell. Using these labels, we compute a Jaccard index (JI) to ascertain how similar are the neighborhoods in terms of cell type composition.

Vesalius demonstrated optimized performance for both cell type score (ARI) and interaction score (JI) in these tests. When measuring the ability to recover the identity of mapped cells in the circle regime, for instance, Vesalius showed comparable performance with SLAT, Scanorama, and CytoSpace (Figure 2a) in ARI. The perfect score of CytoSpace is expected as it only maps cells of the same cell type. With its requirement of shared cell type labels between data sets, CytoSpace is incapable of mapping samples when not all cells are shared (121 were mapped out of 132). This is particularly evident in the dropped regime where CytoSpace only mapped 7 samples out of 132 (see Supplementary Figure S7). We show the mapping performance for all other regimes in Supplementary Figure S6 to S9). We also included a CytoSpace run (CytoSpace_noLab in Figure 2, Supplementary Figure S6-S9) where all cell labels are the same and we see that the performance drops dramatically.

If we consider each tools ability to map spatial context, we see that Vesalius outperforms all other tools by including the right cellular context to construct the cost matrix. Unsurprisingly, including neighborhood composition strongly improves Vesalius’s performance. It is of note that CytoSpace (with true labels) and Scanorama excels at finding the right cell type but fail at recovering context falling behind Vesalius even when it does not include cell composition (e.g. n , fn , nt cost matrices – legend key shown in figure legend). We show pair-wise statistical comparison (Wilcoxon Rank Sum test- see methods) between each tool for ARI scores and Jaccard index scores in Figure 2c and Figure 2d respectively.

We exemplify the performance of each tool by plotting a mapping event in the circle regime (Figure 2e). We aimed to map cells in the query sample onto the reference data sets (panels in the top left of Figure 2e). For clarity, we selected only 3 cost matrix combinations for Vesalius. Specifically, we selected $fncty$, $fnct$ and fnt . Our results indicate that Vesalius is able to effectively map cells from the query sample onto the reference sample while considering spatial context. We noticed that spatial alignment tools such as PASTE and GPSA fail to recover the structure of the reference sample from the query sample. The task of mapping cells across samples is a distinct task from spatial alignment and the constraints applied to the samples by PASTE and GPSA are incapable of overcoming these differences. We show the results for all other synthetic data sets in Supplementary Figures S6 to S9.

Finally, we highlight computational performance (run time and memory usage) when increasing the number of cells from 100 to 200 000 (Supplementary Figure S10) in a synthetic circle regime (See Methods). For each tool, we computed the mean run time and peak RAM usage (Resident Set Size) across 3 runs for each number of cells. We also include a comparison of run time and memory usage across cost matrix combinations for 5000 and 10 000 cells.

Benchmarking mapping performance in real spatial data

We further tested Vesalius using the same 14 cost matrices against other tools (excluding GPSA due to excessive memory requirements) in seqFISH mouse embryo[7], Stereo-seq mouse embryo[8], and Slide-seq V2 mouse hippocampus[9] (see Methods). We found that the mapping by Vesalius with all available information (fncty) showed ARI larger than 0.75 for all dataset we tested. This is compared with other approaches whose performance was compromised when using real data sets. The perfect ARI score for CytoSpace is expected as it only maps to cells with the same cell type. When the cell type information is lost (CytoSpace_noLab), the ARI became drops to less than 0.3 for all data sets. Due to the requirement of matching cell type, CytoSpace cannot run on the Stereo-seq data set without removing label information (Supplementary Figure S11). SLAT, which showed good performances for synthetic data sets showed ARIs around 0.5 and even 0.25 for Slide-seq data. Regardless of the datasets, Vesalius showed the best performance when assessing with JI especially when using cell composition (c). The JI score decreases when adding other configurations to compute the cost in Vesalius while ARI, in general, is enhanced, indicating that Vesalius tries to find a balance between cells and spatial context depending on the configuration.

We show mapping examples using fncty, fnct and fnt. The mapping examples of Vesalius also demonstrate a strong resemblance to the reference (Figure 3c, f and Supplementary Figure S11c). While SLAT's mapping generally aligned with the reference, it exhibited misannotations in highly heterogeneous brain regions (Figure 3f). SLAT and Scanorama still perform decently at correctly mapping cells across seqFISH samples but fall behind when capturing the cellular context. In fact, Vesalius using only cell and niche similarity performs similarly to SLAT at capturing cellular context. Again, PASTE failed to recover the structure of the reference sample from the query sample as observed in the examples using synthetic data.”

Discussion

“In our benchmarking across synthetic and real data sets, we demonstrate how the inclusion multi-scale and multi-context information improves the mapping of cellular spatial context. We observed that tools such as SLAT and Scanorama perform extremely well at accurately mapping cells across synthetic samples (Figure-2a, Supplementary Figure S6-S9). However, they do not capture spatial context with sufficient accuracy to map the cellular and spatial context of cells (Figure 2b). Vesalius, on the other hand, can exploit all available information (including cell type composition of niches and cell type labels) to accurately map cells and their spatial context. It of note that Vesalius showed outstanding performance when dealing with more complicated real data where multiple cell types are co-localized, indicating that Vesalius utilizes spatial context effectively for mapping. In contrast, the high performance of SLAT and Scanorama were not reproduced in real biological data.”

2. Likewise, the authors mainly mention rigid alignment approaches like PASTE. How does Vesalius compare to diffeomorphic (non-rigid) alignment techniques like Spateo (<https://www.biorxiv.org/content/10.1101/2022.12.07.519417v1>), STalign

(<https://www.nature.com/articles/s41467-023-43915-7>),
[cst\(https://www.nature.com/articles/s41592-024-02410-7\)](https://www.nature.com/articles/s41592-024-02410-7), etc?

First, we would like to point out that GPSA provides non-rigid alignment and performs poorly in our benchmarking (See Figure-2 and Supplementary Figures S6 through S9). Tools such as Spateo or STalign also provide non-rigid alignment in the sense that a 2D mesh can be overlaid on the spatial data and can be differentially and locally deformed. However, this mesh can never be broken, folded or scrambled. These tools assume that the relationship between tissue structures will be shared between samples. Moreover, STalign uses manually selected tissue landmarks to serve as anchors for diffeomorphic alignment. This assumption of structural rigidity may hold for adjacent tissue section or highly structured tissue such as the brain. It does not, however, hold for highly heterogenous tissue structures such as the ones shown in our synthetic data, cancer data sets (Prostate Cancer and IMC), and seqFISH embryo data (See Figure-2, Supplementary Figure S1-S9, Figure-3a/c, Figure-4, Figure-5, Supplementary Figure S15)

We mentioned PASTE and GPSA to clearly demonstrate that mapping cells across samples is a distinct task to the one of spatial alignment. As Spateo and STalign share the same assumption as PASTE and GPSA, the results will be similar. As seen in our benchmarking, adding cell interaction (e.g. the cellular niche or cell type composition) is extremely useful in mapping spatial data. However, cell interaction information is never used for PASTE, GPSA, Spateo and STalign. In our synthetic data, we see that GPSA and PASTE merely rotate the query sample. The structure of the query sample is not modified at all. This is not the case of tools such as Vesalius, SLAT, Scanorama, and Tangram which map individual cells across samples (See Figure-2, Supplementary Figure 6 through 9). It is for these reasons that these tools can recover the spatial structure of reference using the cells from the query. The flexibility of Vesalius allows us to achieve this much more accurately than other tools and with much higher flexibility catering to the diverse needs of the scientific community.

We clarified our intent by adding a section in the introduction and discussion concerning spatial alignment and how this is indeed a separate task than spatial alignment.

Figure 2a/b: Performance benchmarking in synthetic circle regime. PASTE and GPSA perform poorly at mapping cells across samples since they perform the distinct task of spatial alignment.

Figure 2e: The goal is to map the cells of the query onto the reference. GPSA and PASTE maintain the structure of the query and attempt to rotate or distort the sample to fit on the reference. However, given the structural heterogeneity, they both fail at recovering the structure of the reference.

Changes to the Manuscript

Introduction

“And yet, we have seen the emergence of 3D spatial stacks[8, 10, 11] taken from adjacent tissues and, in parallel, a plethora of tools allowing the alignment of these tissue slices including PASTE[3], GPSA[4], or STalign[12] have been developed. The assumption made by these tools is that adjacent sections share sufficient structural similarity to locally and differentially deform tissue sections to match the neighboring section. Importantly, these methods do not fold, break or scramble the cells present in the tissue rather they displace them by maintaining their relative position to each other”

Discussion

“Mapping cells across heterogeneous samples is a distinct task from spatial alignment of spatial samples and should not be confused. Tools such as PASTE[3], GPSA[4] or STalign[12] make an assumption of structural rigidity which is expected in adjacent tissue samples. While they can locally and differentially deform a query sample to match a target sample, the query sample cannot be broken, folded, or scrambled. However, samples taken from a tumor (even if it is the same tumor) are going to be structurally heterogeneous and this heterogeneity will be further amplified when sampling across patients. We demonstrate in our benchmarking (synthetic and real data) that spatial alignment tools (PASTE and GPSA) are incapable of mapping cells between samples in a way that will recover the structure of morphologically complex tissues.”

3. In general, the accuracy of the alignment on real spatial transcriptomics data was unclear. The authors need to include more quantitative evaluations.

Following the reviewer's suggestion, we have included further quantitative benchmarking of Vesalius across 3 different technologies (seqFISH, Stereo-seq, and Slide-seq) using a variety of cost matrices. We selected these data sets since they offer a balance between structural heterogeneity (especially in the case of seqFISH mouse embryo data) and they increase the ease with which one can intuitively judge the performance of each tool.

Nevertheless, we quantitatively demonstrate the performance of Vesalius compared to SLAT, Scanorama, CytoSpace, Tangram, and PASTE. We excluded GPSA due to excessive run time and memory requirement for larger data sets. We had set a hard memory limit of 480GB which GPSA consistently exceeded. Similarity to our synthetic benchmarking, we measure the ability of each tool to capture the cell type of matched cells and the spatial context of matched cells. In **figure 3**, we show performance and mapping events for seqFISH mouse embryo and Slide-Seq V2 mouse hippocampus. In **Supplementary Figure S11**, we show the mapping performance in Stereo-Seq mouse embryo data.

Once again, CytoSpace achieve perfect ARI score due to the fact that it only maps cells between cell types. Its performance is particularly poor when cell labels are not provided. Furthermore, in Stereo-seq, there are unshared cell labels between mouse embryo sections which CytoSpace was unable to handle.

Vesalius, on the other hand, not only captures the cell type of the center cell but also accurately capture the context of the cell. The ability to flexibly use different cost matrices allows a user to define a mapping strategy that best fits their research question. The flexibility of this our strategy also allows to deal with potential cell type mislabeling which can occur when using cell type annotation tools.

Changes to the Manuscript

Results

“Benchmarking mapping performance in real spatial data

We further tested Vesalius using the same 14 cost matrices against other tools (excluding GPSA due to excessive memory requirements) in seqFISH mouse embryo[7], Stereo-seq mouse embryo[8], and Slide-seq V2 mouse hippocampus[9] (see Methods). We found that the mapping by Vesalius with all available information (fncty) showed ARI larger than 0.75 for all dataset we tested. This is compared with other approaches whose performance was compromised when using real data sets. The perfect ARI score for CytoSpace is expected as it only maps to cells with the same cell type. When the cell type information is lost (CytoSpace_noLab), the ARI became drops to less than 0.3 for all data sets. Due to the requirement of matching cell type, CytoSpace cannot run on the Stereo-seq data set without removing label information (Supplementary Figure S11). SLAT, which showed good performances for synthetic data sets showed ARIs around 0.5 and even 0.25 for Slide-seq data. Regardless of the datasets, Vesalius showed the best performance when assessing with JI especially when using cell composition (c). The JI score decreases when

adding other configurations to compute the cost in Vesalius while ARI, in general, is enhanced, indicating that Vesalius tries to find a balance between cells and spatial context depending on the configuration.

We show mapping examples using fnc_{cty}, fnc_t and fnc_t. The mapping examples of Vesalius also demonstrate a strong resemblance to the reference (Figure 3c, f and Supplementary Figure S11c). While SLAT's mapping generally aligned with the reference, it exhibited misannotations in highly heterogeneous brain regions (Figure 3f). SLAT and Scanorama still perform decently at correctly mapping cells across seqFISH samples but fall behind when capturing the cellular context. In fact, Vesalius using only cell and niche similarity performs similarly to SLAT at capturing cellular context. Again, PASTE failed to recover the structure of the reference sample from the query sample as observed in the examples using synthetic data.”

Figure-3: Benchmarking mapping performance in real spatial data. (a) ARI of mapped cell type labels between a reference seqFISH mouse embryo data set and a query seqFISH mouse embryo data set. We compared Vesalius using 14 different cost matrices to SLAT, CytoSpace, CytoSpace with no cell type labels (noLab), Tangram, Scanorama, and PASTE. In seqFISH data, the addition of cell type labels or composition of cellular neighborhoods improves Vesalius's ability to accurately map cells across samples even outperforming SLAT and Scanorama. **(b)** JI of cell interactions of the neighborhood ($k=10$). **(c)** Example mapping event in seqFISH mouse embryo. We show mapping events for 3 Vesalius cost matrices (fntcy, fntct, fnt). **(d)** ARI of mapped cell type labels between a reference Slide-seq V2 mouse hippocampus data set and a query Slide-seq V2 mouse hippocampus data set. **(e)** JI of cell interactions of the neighborhood ($k=10$). **(f)** Example mapping event in Slide-seq V2 mouse hippocampus. We show mapping events for 3 Vesalius cost matrices (fntcy, fntct, fnt)

4. The Vesalius algorithm appears to scale as $O(n^3)$ and $O(n^4)$, which is highly inefficiency and would take a very long time to run, especially given newer higher resolution and generally larger datasets. How long does it take to run as a function of the number of cells and labels, niches, territories, etc, evaluated?

We agree with the reviewers that performance is an important consideration especially for high resolution data sets. We also realize that the description of the core algorithm was not sufficiently clear to highlight Vesalius's strategy towards performance. The LAPVJ algorithm ($O(n^3)$) is an improved version of the Hungarian mapping strategy ($O(n^4)$). We only use the LAPVJ algorithm in Vesalius. We made it clear in the revised manuscript.

In Vesalius, the performance of the mapping will be affected by the number of cells and the number of cost matrices we compute. We compute a cost for each individual cell which can also include the average expression profile of its local neighborhood or large-scale tissue territories. Correlation scores are assigned to each cell individually.

We improve the performance of the mapping by dispatching cells into batches which can be scored and mapped in parallel. At every epoch, we ensure that all cells are selected at least once for mapping. The final mapping results represents the cells which showed the lowest overall cost across all batches and epochs.

With this in mind, we included a benchmarking for computational performance by varying the number of cells used during mapping (100 to 200k). For this task, we aimed at making the performance as fair as possible as such we assumed that a user would run a full analysis pipeline including data loading, preprocessing, and mapping on a single CPU. We made this assumption since all tools have different input requirements, different downstream tasks, and different optimization options. Vesalius provides the widest variety of potential downstream task and as such has the most requirement in terms of pre-processing. This strategy puts our tool at a disadvantage but it was the least objectionable way to benchmark performance. We use the same strategy to compare run time across cost matrix combinations (using 5000 and 10000 cells).

We also aimed to avoid any variation in language specific methods of measuring run time and memory usage. We measured run time and memory (Resident Set Size – peak RAM usage) for each tool using custom bash scripts (Available on the dedicated GitHub page) that measured these metrics across 3 submitted jobs. We set hard computational limits: a maximum of 480GB of RAM and maximum run time of 12 hours. If the job exceeded these limits, the run was cut short. Despite its higher pre-processing requirements, Vesalius strikes a balance between mapping performance, computational performance and interpretability. We show computational performance in **Supplementary Figure S10**.

Changes to the Manuscript

Results

“Finally, we highlight computational performance (run time and memory usage) when increasing the number of cells from 100 to 200 000 (Supplementary Figure S10) in a synthetic circle regime (See Methods). For each tool, we computed the mean run time and peak RAM usage (Resident

Set Size) across 3 runs for each number of cells. We also include a comparison of run time and memory usage across cost matrix combinations for 5000 and 10 000 cells. “

Methods

“In addition to these data sets, we also generated 3 samples of the circle regime with varying number of cells (100, 500, 1000, 2500, 5000, 10000, 20000, 50000, 100000, 200000). These data sets were used for computational performance benchmarking (see below).”

“Computational performance benchmarking

We included a benchmarking for computational performance by varying the number of cells used during mapping (100, 500, 1000, 2500, 5000, 10000, 20000, 50000, 100000, 200000) in a circle synthetic regime.

For this task, we aimed at making the performance as fair as possible as such we assumed that a user would run a full analysis pipeline including data loading, preprocessing, and mapping on a single CPU. We made this assumption since all tools have different input requirements, different downstream tasks, and different optimization options. Vesalius provides the widest variety of potential downstream task and as such has the most requirement in terms of pre-processing.

We also aimed to avoid any variation in language specific methods of measuring run time and memory usage. We measured run time and memory usage (Resident Set Size – peak RAM usage) for each tool using custom bash scripts (Available on the dedicated GitHub page) that measured these metrics across 3 submitted jobs. We set hard computational limits: a maximum of 480GB of RAM and maximum run time of 12 hours. If the job exceeded these limits, the run was cut short.

We compared run time and memory usage across cost matrix combinations for 5000 and 10 000 cells. We used the same 14 matrix combinations described above and used the same performance measuring strategy as described above.

”

present technical heterogeneity (differences in resolution – Figure 4c/d). The IMC data sets used in this study were taken from hundreds breast cancer patients where we investigate multi-scale and multi-context patterns trends of cells states across patients. We believe that our results in Figure-2, Supplementary Figure 1-9, Figure-3a/c, Figure-4, Figure-5, Supplementary Figure-S15 demonstrate sufficient spatial heterogeneity.

To further demonstrate the mapping ability of our algorithm, we included mapping of prostate cancer tumors using Slide-seq V2 technology. Our results show that despite structural heterogeneity our method is able to map cells across tumor samples (See Supplementary Figure S15).

Changes to the Manuscript

Results

“To demonstrate that our mapping strategy is able to map cells across highly heterogeneous tissues such as cancer, we mapped cells from prostate cancer tumor samples using Slide-seq V2 technology. Using the meta data provided by the authors, we mapped cells across samples using cell, niche, territory, and cell type labels to construct our cost matrices. We show a mapping example in Supplementary Figure S15. Our results indicate that even in highly heterogenous cancer samples, Vesalius is able to map cells across samples. “

Methods

“Mapping across spatial heterogenous prostate cancer samples

We mapped cells from 2 prostate cancer samples using Slide-seq V2 technology[13]. In this instance, we downloaded the count matrices, spatial barcodes, and cell type annotations provided by the authors. We processed 2 tumor samples (Tumor01 and Tumor02) using the Vesalius pipeline. In brief, we normalized gene counts, selected variable features ($n = 2000$), and reduced dimensionality via PCA (30 PCs). Next, we smoothed the Vesalius image stacks and equalized the color histogram before segmenting the images and isolating territories. We mapped each sample to each other using cell, niche (KNN with $k = 15$), territory similarities, and cell type labels. The mapping procedure was carried out across 25 epochs with a batch size set to 5000 consistent with our previous analysis. The details of the analysis are available on the dedicated GitHub. “

Figure-S15: Mapping of cells across prostate cancer Slide-seq V2 data. We mapped cells from the query data set onto the reference. Mapped cells are shown in the mapped panel. The mapping results indicate that Vesalius recovers the location of cells as well as the location of the tumor.

6. The website link is broken <https://patrickcnmartin.github.io/Vesalius/>

We thank the reviewers for picking up on this oversight. The package had been migrated to a different GitHub repository and the link to the website had not been correctly updated.

New link: <https://wonlab-cs.github.io/Vesalius/>

7. The vignettes while existent are not compiled, making it difficult for us to evaluate whether the code is functional

The vignette has been compiled on the new web page and can be found here:

<https://wonlab-cs.github.io/Vesalius/articles/vesalius.html>

To maintain a light-weight web page, the analysis presented in this manuscript is available on a dedicated GitHub page. The README presents an overview of the analysis.

https://github.com/WonLab-CS/Vesalius_analysis

Reviewer 2

This paper proposes a computational tool developed to align spatial data, from the same technology, from adjacent tissue sections, resulting in effective cell, tissue and niche mapping strategies. Additionally, the tool demonstrated relatively effective mapping between different spatial technology assays (Stereo-seq and seqFISH) and different assay resolutions (Visium and Visium HD).

The rapid rise of spatial transcriptomic technologies has resulted in large amounts of complex data generated with few analysis methods tailored to the type of data produced. Creating novel analysis tools such as this cell mapping method, are essential if the spatial transcriptomics field is to last. In particular, methods focused on alignment or integration are currently rare, though the need for this is growing, making the significance of this paper important. This paper claims to be able to maps cells across adjacent tissue sections, which it succeeds in doing so, recovering tissue structures within brain and embryo samples. It may be of interest to test this tool of less structurally sound datasets such as cancer samples. However the current trends when developing any spatial transcriptomic associated research is to test on brain samples, the samples selected are well within the standards of the field.

A few suggestions would be;

1. Include some statistics along with figure 2a boxplots when comparing the different method performances.

For clarity, we do not include statistics within the box plots but rather show significance levels as heat maps. For each tool comparison, we performed a Wilcoxon Rank Sum Test and plot the p-value. We chose a Rank sum test due to CytoSpace having 0 variance in the ARI scores. This ensures that we remain consistent in our comparisons across all tools. We also compute the significance levels for our measure of cell context. For each sample, we added the cell labels for each spatial coordinate as well as the labels of the 6 nearest neighbors (“interaction” column in the synthetic data produced by oneiric). We can compute a Jaccard Index between these interactions labels to gage how well each tool is able to capture spatial context. We compute the significance levels using the same strategy.

We have added these heat maps for all synthetic regimes, and these results are shown in Figure-2 and Supplementary Figures S6 through S9.

Figure 2c/d: statistical significance between tools and cost matrices in the synthetic circle regime
Changes to the manuscript

Results

"Benchmarking mapping performance in synthetic spatial data"

We show pair-wise statistical comparison (Wilcoxon Rank Sum test- see methods) between each tool for ARI scores and Jaccard index scores in Figure 2c and Figure 2d respectively."

2. Is there some type of mapping score in figure 3.e that could be done to test between the different technologies?

We agree with the reviewers that this would indeed be an interesting metric to measure for figure 3.e (currently **Figure 4a/b**). Unfortunately, there is no straightforward way of scoring the mapping between technologies. One approach is to re-annotate both data sets using the same labels. In fact, this is what we had done for the VisiumHD to Visium mapping: both data sets were annotated using the same reference scRNA data with RCTD[14]. However, optimal annotation methods will differ between technologies and is a time-consuming task especially without domain experts to guide cluster annotations. As such, we preferred using the expert annotations provided by the original authors when available.

To showcase how cells are being mapped across samples, we have added an alluvial plot showing where each cell is being mapped to across data sets (See **Figure 4b**). We have also included a measure of accuracy (0.901) of global tissue labels such as "brain" and "Notochord" for stereo-seq and "Forebrain/Midbrain/Hindbrain", "Spinal cord" for seqFISH.

Changes to the manuscript

Results

"More often than not, data sets of interest are generated using different technologies and yet we would still like to mine insights from them. We tackled this challenge using our flexible mapping strategy. First, we mapped high-resolution Stereo-seq mouse embryo[8] (E9.5) onto image-based seqFISH[7] in the same tissue at a slightly earlier developmental stage (E8.75). While both data sets contain cell type labels, these differ greatly between data sets as is often the case when using data from different sources. As such we forwent the use of cell type labels, and we constructed the total cost from cell similarity and niche similarity only. Overall, the broad cell type labels matched well when mapping across technologies (Figure 4a). We show the mapping of cells to different cell labels in an alluvial plot (Figure 4b). For instance, Brain and Notochord cells in the Stereo-seq are mapped to Forebrain/Midbrain/Hindbrain and Spinal Cord in seqFISH with 0.901 accuracy. We show the cell mapping results for Stereo-seq to all 3 seqFISH sections in Supplementary Figure S13. We also performed the converse mapping - seqFISH mapped to Stereo-seq - in Supplementary Figure S14."

Figure-4: Cell Mapping across technologies and resolutions. (a) Mapping of Stereo-seq mouse embryo cells onto seqFISH mouse embryo cells shows that Vesalius recovers expected tissue structures across technologies. **(b)** Alluvial plot demonstrating where each mapped cell types in the query (Stereo-seq) maps to in the reference (seqFISH). Despite different cell type labels major cell type are mapped to their expected counterpart.

3. Colour legend is missing from a few of the figures (figure 3.c and 3.f).

We have added the color legends when appropriate. In the new figures, we have added the cell type labels instead of Vesalius Territories. **Changes made to Figure 3 and Supplementary Figure S11.**

4. When looking at Visium and Visium HD cell type spot overlap (figure 3.f), a percentage of spots with a high Jaccard index would help clarify and strengthen the statement 'most spots contain the same cell types between technologies' made. E.g. 70% of spots have an index of 0.7 and above.

We thank the reviewer for this comment. We had made a mistake in the text and the plot originally shown in figure 3f (now shown in **Figure 4d**) was not a Jaccard index of cell type overlaps. We had reasoned that cell type proportions should play a crucial role in how we assess cross resolution performance. Using the cell type proportions provided by RCTD and the cell labels mapped to each spot, we performed a chi-squared test to ascertain if the frequency of cells was statistically different between Visium spots and mapped VisiumHD cells. The plot showed the p-value for each spot. Nearly every spot showed that there was no significant difference in cell type frequencies. (See **Figure 4d**). We also show a discretized version of the p-values highlighting that only a few spots showed statistically different cell type frequencies. However, we also included the Jaccard index of mapped cells as well as the distribution of scores across all spots. The distribution includes vertical lines highlighting 25th, 50th and 75th quantiles (**Figure 4e**).

Changes to the manuscript Results

“To further investigate the cell types matching abilities between resolutions, we used RCTD[14] to estimate cell type proportions in both Visium and VisiumHD using the same reference scRNA-seq data set[15]. We scaled cell type frequencies (see methods) to create a contingency matrix of cell type proportions. We performed a Chi-squared test to determine wherever cell type proportions exhibited significant differences between Visium and mapped VisiumHD. The p-value distributions across all mapped cells show that nearly all mapped cells display similar cell type proportions (**Figure 4d**). In fact, when we discretized the p-value, we observed only a few instances where cell type frequencies statistically differed. In addition, we computed a Jaccard index between cell types associated with Visium spots and the mapped cell

identities from VisiumHD. Despite non-significant difference in cell type proportions, we wanted to see to what extent are cell type labels missed during mapping. Our results show a reasonable correspondence in mapped cell types between Visium and VisiumHD (Figure 4e). The distribution of Jaccard scores show that the median Jaccard score is around 0.32 (Figure 4e - left panel with 25th, 50th and 75th percentile shown by vertical orange, blue and green lines). Taken together, our results indicate that while Vesalius will occasionally miss cell types associated with Visium spots or VisiumHD cells, the cell type frequencies do not statistically differ.”

Figure-4: Cell Mapping across technologies and resolutions. (d) We show the mapped p-values after performing a Chi-squared test between cell type frequencies in Visium (after cell type cell type deconvolution) and the mapped cell types taken from VisiumHD. A discretized p-value map shows that only a few spots exhibit a statistically different cell type frequency between spots and mapped cells. **(e)** Jaccard Index of mapped cell labels between Visium and VisiumHD. The distribution of Jaccard Index scores show that 50% of cells have a score above 0.32. Orange vertical line represents 25th percentile. Light blue line represents 50th percentile. Green line represents 75th percentile

Methods

“To compare the cell types mapped across resolution, we first scaled the cell type frequencies in the deconvoluted Visium Spots to assume 100 total cells in each spot. We applied the same frequency scaling to the cell type mapped from Visium HD. Using these scaled cell type frequencies, we constructed a contingency matrix upon which we applied a Chi-squared test to determine if the cell type frequencies were statistically distinct. In addition, we computed a Jaccard Index between cell type labels in the Visium spots and mapped VisiumHD cells to determine how many cell types were being missed despite having potentially low occurrence frequencies.”

5. In figure 4 (a and d) it would be beneficial to have the labels for 'reference', 'mapped' etc.

We have amended the figure as requested.

Changes to the manuscript Figures

Figure-5: Cell Mapping across time. (a) Mapping Stereo-seq Axolotl brain regeneration data sets (ARTISTA) at 15 days post injury (DPI) to 20 DPI using cell, niche, and territory similarity. Blue circle highlights ependymogial cells and their transition to regenerative intermediate progenitor cells. (b) Integrated tissue territories of mapped cells and reference cells where tissue territories are recovered using a shared latent space. For visualization purposes, we separate both samples. (c) Two differentially expressed genes at the zone of injury (territory 8 – dark blue): AMEX60DD048805 and AMEX60DD022398 show expression patterns limited to a small subset of cells involved in cell differentiation and proliferation necessary for wound healing. (d) Mapping Stereo-seq Mouse embryo (MOSTA) development at stage E11.5 to stage E12.5 using cell, niche and territory similarity. (e) Clustering of query brain cells mapped onto the reference brain cells. Brain cells are demarcated into sub-territories including cells that lay at the interface between brain regions (cluster 2) and the interface between the brain and other tissues (clusters 4). (f)

Differential gene expression between clusters exemplified by Hes5, Stnm2, and Cancng4 as genes highlighting the different spatial sub-structures present in the mouse brain and the accurate mapping of gene expression patterns across time. (g) Crabp1 – a gene involved in the regulation of stem cell differentiation – shows a spatial-temporal decoupling behavior where the expression shows a spatial demarcation at the earlier developmental stage (E11.5) which is then extinguished at the following developmental stage.

6. Figure 5c is not referred to in the text, even though the results are discussed which initially led to some confusion.

We are grateful to the reviewer for picking up on this. It was indeed an oversight which we have now fixed. The figure is now reference as figure 6c in the manuscript.

Changes to the manuscript

Results

“We further performed hierarchical clustering of the ER negative patients (n = 50) using our total cost as a measure of distance between samples (Figure 6c). In addition to cost, we show the average performance associated with each cost metric. “

Reviewer 3

The authors have developed a tool previously that is able to identify spatial structures in spatial molecular data. The manuscript here is an extension, in which the authors leverage the so-called territories, to integrate spatial molecular data with a query dataset. For this task, the authors implement a linear assignment strategy as part of the previous package. The authors suggest that this strategy outperforms competing strategies.

General comments

Personally, I find this work exciting and looking forward to its publication. The phrase “I think there is something here” keeps popping into my head. It just needs to be cleanly worked out and shown. I find it crucial to assess it more carefully against other approaches, such that it also becomes clear towards the reader and user, in which situations this approach is going to be superior, and in which cases others maybe preferable. Computing and memory requirements is a major concern with these approaches, but also proper benchmarking should be done to provide a more complete picture. I am actually willing to believe that a LAP solution may beat a badly trained NN, but formally I like to see the evidence (see more detailed comments).

1. Most evaluation is qualitative. Only limited quantitative benchmarks. I like to see more quantitative evaluations. There are already benchmark papers out there (e.g.<https://doi.org/10.1038/s41592-022-01480>). I generally do not believe that a novel approach needs to beat all tools in all scenarios to be great pieces of new research (in case this was the motivation not to systematically assess it).

We appreciate the reviewer’s flexibility on the matter of benchmarking. To provide a quantitative measure, we provide more scenarios in synthetic data with additional quantitative measures. We

provide quantitative measure for 3 real biological datasets namely seqFISH, Slide-seq and Stereo-seq.

We modified the synthetic data to provide increased complexity to the samples we would map. Tissue territories can be of different shapes and sizes. These territories may contain multiple cell type or even layers. We also modified our quantitative assessment strategy. We realized that using Adjusted Rand Index (ARI) on cell type labels was not an optimal strategy since it did not account for spatial context explicitly. As such, we created new cell type labels which reflect local cell-cell interactions. Specifically, this label captures the cell types of 6 (or 10 on real data) nearest neighbors (defined by K-NN). Using these labels, we can compute a Jaccard Index (JI) to assess if we are indeed capturing spatial context. The assumption is that an accurate mapping should not only recover the cell type of matched cells but the cell types of the neighboring cells (i.e. the spatial context). We applied the same principle to 3 real biological data sets namely seqFISH, Slide-seqV2 and Stereo-seq. For each data set, we measure the ability to capture the center cell (ARI) as well as the composition of the neighborhood (JI).

In both cases, we show the performance of Vesalius using different cost matrix combinations and how these combinations will affect the ability to recover the label of the center cell and the spatial context. Since the synthetic data offers 132 mapping events (12 samples mapped to each other excluding self-mapping), we show the results as box plots and provide a heat map showing statistical significance between tools and cost matrix combinations for Vesalius. We do not provide these metrics for the real data since there was only one example in each case.

Quantitative assessment of Vesalius and cost matrix combinations in comparison to other tools (SLAT, Scanorama, CytoSpace, Tangram, PASTE, GPSA) are shown in Figure 2, Figure 3, Supplementary Figure S6 through S9, and Supplementary Figure S11.

We have also included computational performance benchmarking which is addressed below.

Changes to Manuscript

Results

“Benchmarking mapping performance in synthetic spatial data

To demonstrate the effectiveness of our mapping strategy, we elected to use synthetic spatial data since this will provide a robust ground truth (See methods). Specifically, we aimed to demonstrate our ability to accurately map cells across samples along with their spatial context. First, we generated 5 synthetic spatial regimes (circle, layer, dropped, random one cell, random two cell) that mimic the complexity of biological scenarios (Supplementary Figures S1-S5). For each regime, we generated 12 samples where we aimed to map a query data set on to a reference data for a total of 132 mapping events (excluding 12 self-mapping events). Details of each regime are developed in the method section and each sample can be visualized in Supplementary Figures S1-S5.

Since Vesalius provides a flexible framework to construct the total mapping cost, we compared the performance of Vesalius across 14 different cost matrices in addition to two scRNA-to-spatial mapping tools (CytoSpace[2] and Tangram[2]), spatial alignment tools (PASTE[3], and GPSA[4]) and scRNA/Spatial data integration tool such Scanorama[1] and SLAT[5]. The cost matrix combinations are built using cell similarity (f = feature), niche similarity (n = niche), territory similarity (t = territory), niche cell type composition (c = composition), cell type labels (y = cell type labels) or a combination of these metrics.

To measure the ability of each tool to effectively map cells across samples, we computed an Adjusted Rand Index (ARI)[6] between mapped cell labels. In parallel, we measured each tool's ability to capture spatial context by adding an "interaction" label to our data sets (see Methods). This label concatenates the cell type labels of k nearest neighbors ($k = 6$) of a center cell. Using these labels, we compute a Jaccard index (JI) to ascertain how similar are the neighborhoods in terms of cell type composition.

Vesalius demonstrated optimized performance for both cell type score (ARI) and interaction score (JI) in these tests. When measuring the ability to recover the identity of mapped cells in the circle regime, for instance, Vesalius showed comparable performance with SLAT, Scanorama, and CytoSpace (Figure 2a) in ARI. The perfect score of CytoSpace is expected as it only maps cells of the same cell type. With its requirement of shared cell type labels between data sets, CytoSpace is incapable of mapping samples when not all cells are shared (121 were mapped out of 132). This is particularly evident in the dropped regime where CytoSpace only mapped 7 samples out of 132 (see Supplementary Figure S7). We show the mapping performance for all other regimes in Supplementary Figure S6 to S9). We also included a CytoSpace run (CytoSpace_noLab in Figure 2, Supplementary Figure S6-S9) where all cell labels are the same and we see that the performance drops dramatically.

If we consider each tools ability to map spatial context, we see that Vesalius outperforms all other tools by including the right cellular context to construct the cost matrix. Unsurprisingly, including neighborhood composition strongly improves Vesalius's performance. It is of note that CytoSpace (with true labels) and Scanorama excels at finding the right cell type but fail at recovering context falling behind Vesalius even when it does not include cell composition (e.g. n , fn , nt cost matrices – legend key shown in figure legend). We show pair-wise statistical comparison (Wilcoxon Rank Sum test- see methods) between each tool for ARI scores and Jaccard index scores in Figure 2c and Figure 2d respectively.

We exemplify the performance of each tool by plotting a mapping event in the circle regime (Figure 2e). We aimed to map cells in the query sample onto the reference data sets (panels in the top left of Figure 2e). For clarity, we selected only 3 cost matrix combinations for Vesalius. Specifically, we selected $fnct_y$, $fnct$ and fn_t . Our results indicate that Vesalius is able to effectively map cells from the query sample onto the reference sample while considering spatial context. We noticed that spatial alignment tools such as PASTE and GPSA fail to recover the structure of the reference sample from the query sample. The task of mapping cells across samples is a distinct task from spatial alignment and the constraints applied to the samples by PASTE and GPSA are incapable of overcoming

these differences. We show the results for all other synthetic data sets in Supplementary Figures S6 to S9.

Finally, we highlight computational performance (run time and memory usage) when increasing the number of cells from 100 to 200 000 (Supplementary Figure S10) in a synthetic circle regime (See Methods). For each tool, we computed the mean run time and peak RAM usage (Resident Set Size) across 3 runs for each number of cells. We also include a comparison of run time and memory usage across cost matrix combinations for 5000 and 10 000 cells.

Figure-2: Benchmarking mapping performance in synthetic spatial data – circle regime. We provide a Legend Key on the figure describing the nomenclature of Vesalius cost matrices as well as the meaning of acronyms. **(a)** Performance comparison in recovering cell types across samples using an ARI in synthetic spatial data. We compared Vesalius using 14 different cost matrices to SLAT, CytoSpace, CytoSpace with no cell type labels (noLab), Tangram, Scanorama, PASTE, and GPSA. Number of samples used in each box plots included in axis labels. Median line and interquartile range shown in each boxplot (center line and whiskers). CytoSpace obtains a perfect score since it only maps cells from the same cell type. **(b)** JI of cell interactions. To account for the

spatial context of cells, we computed the JI between cell type labels in the neighborhood of mapped cells ($k=6$). **(c)** Heat map representing p -values after performing Wilcoxon rank sum test between tools on ARI scores. **(d)** Heat map representing p -values after performing Wilcoxon rank sum test between tools on JI scores. **(e)** Example mapping event in the circle regime. In the circle regime, there are 5 randomly sized and randomly placed territories placed in a background territory. Each circle can contain 2 different cell types. We show mapping events for 3 Vesalius cost matrices (fncty, fnct, fnt).

Methods

“Benchmarking mapping performance

We contrast our mapping approach to single cell mapping tools, spatial alignment tools, and data integration tools. More specifically, we compared Vesalius to CytoSpace[16], Tangram[16], PASTE[3], GPSA[4], SLAT[5], and Scanorama[1]. It is of note that CytoSpace requires perfect matching of cell type labels between reference and query and only finds a proposed mapping of cells between the same cell type labels. Importantly, if the cell labels are not shared CytoSpace will not run to completion. To demonstrate how this strategy differs from our context specific approach to cell mapping, we provided CytoSpace with true cell labels as well as a single cell label for all cells (noted with the `_noLab` label).

For Vesalius, we ran mapping with 14 cost matrix combinations:

- feature
- niche
- territory
- composition
- feature-niche
- feature-niche-cell_type
- feature-composition
- feature-territory
- niche-territory
- niche-composition
- feature-niche-territory,
- feature-niche-composition
- feature-niche-composition-territory
- feature-niche-composition-territory-cell_type

Where features represent cell similarity, niche represents niche similarity, territory represents territory similarity, composition represents cell type composition of niche, and cell type represent the cell label score.

To quantitatively assess the performance of cell mapping between samples, we computed an Adjusted Rand Index[6] (ARI) between the cell labels of the query and the reference. We also computed a Jaccard Index between cell labels of the neighborhood of mapped cells to ascertain if we were capturing the spatial and cellular context of each mapped cell. Since PASTE and GPSA do not match cells but rather modify the coordinates of each cell, we computed the nearest cell neighbor between reference and query and compared their cell type labels. While this approach disadvantages PASTE and GPSA, our objective is to demonstrate that context mapping is a fundamentally different task than spatial alignment. Scanorama is a tool designed to integrate single cell data sets; it only provides an integrated latent space but not best

matching cells. As such, we computed the nearest neighbors between the reference data and query data in the integrated latent space to determine best matching cell.

For Vesalius, we ran our mapping using 1000 cells (for synthetic data) or 5000 cells (for real data) across 25 epochs. We set a cell filtering threshold of 0.9 and allowed cell type label filtering if they were provided. We used the log normalized counts to define gene expression signal. Niches were defined using KNN with $k = 6$ for synthetic data and $k = 10$ for real data. In both cases, we select the top 2000 variable genes for analysis.

2. Figure 2 contains comparisons to existing approaches only on simulated data. While comparison in a controlled dataset makes sense I would have liked to see performance differences on the real datasets as well.

We applied the scoring method described above to 3 different spatial technologies namely Slide-seq, Stereo-seq and seqFISH. However, we dropped GPSA from the analysis in real data since we were unable to run this method to completion due to excessive memory requirement (>480GB RAM) and run times (> 12 hours). Our results indicate that Vesalius perform significantly better in real data than in the synthetic data compared to other tools. In addition, we show a mapping example across all 3 technologies (Figure 3 and Supplementary Figure S11).

Changes to the manuscript

Results

“Benchmarking mapping performance in real spatial data

We further tested Vesalius using the same 14 cost matrices against other tools (excluding GPSA due to excessive memory requirements) in seqFISH mouse embryo[7], Stereo-seq mouse embryo[8], and Slide-seq V2 mouse hippocampus[9] (see Methods). We found that the mapping by Vesalius with all available information (fncty) showed ARI larger than 0.75 for all dataset we tested. This is compared with other approaches whose performance was compromised when using real data sets. The perfect ARI score for CytoSpace is expected as it only maps to cells with the same cell type. When the cell type information is lost (CytoSpace_noLab), the ARI became drops to less than 0.3 for all data sets. Due to the requirement of matching cell type, CytoSpace cannot run on the Stereo-seq data set without removing label information (Supplementary Figure S11). SLAT, which showed good performances for synthetic data sets showed ARIs around 0.5 and even 0.25 for Slide-seq data. Regardless of the datasets, Vesalius showed the best performance when assessing with JI especially when using cell composition (c). The JI score decreases when adding other configurations to compute the cost in Vesalius while ARI, in general, is enhanced, indicating that Vesalius tries to find a balance between cells and spatial context depending on the configuration.

We show mapping examples using fncty, fnct and fnt. The mapping examples of Vesalius also demonstrate a strong resemblance to the reference (Figure 3c, f and Supplementary Figure S11c). While SLAT’s mapping generally aligned with the reference, it exhibited misannotations in highly heterogeneous brain regions (Figure 3f). SLAT and Scanorama still perform decently at correctly mapping cells across seqFISH samples but fall behind when capturing the cellular context. In fact, Vesalius using only cell and niche similarity

performs similarly to SLAT at capturing cellular context. Again, PASTE failed to recover the structure of the reference sample from the query sample as observed in the examples using synthetic data.

Finally, we further investigated the role played by each cost matrix in mapping cells across samples. We first computed a coefficient of variation (CV) on each cost matrix across these 3 biological samples (seqFISH, Stereo-seq, and Slide-seq V2). We reasoned that a higher CV would indicate a higher ability for a cost matrix to discriminate cells during the mapping. We also computed a Proportion of Contribution (POC) which measure the proportion of contribution of each cost matrix to the total cost used for mapping (Supplementary Figure S12). A higher POC is associated with higher JI or correlation. Cell type labels and neighborhood composition show a higher ability to discriminate cells during mapping but participate much less in the total mapping cost. On the contrary, territory and niche similarity tend to have a low CV but participate significantly to the total cost of mapping.

Figure-3: Benchmarking mapping performance in real spatial data. (a) ARI of mapped cell type labels between a reference seqFISH mouse embryo data set and a query seqFISH mouse embryo data set. We compared Vesalius using 14 different cost matrices to SLAT, CytoSpace, CytoSpace with no cell type labels (noLab), Tangram, Scanorama, and PASTE. In seqFISH data, the addition of cell type labels or composition of cellular neighborhoods improves Vesalius's ability to accurately map cells across samples even outperforming SLAT and Scanorama. **(b)** JI of cell interactions of the neighborhood ($k=10$). **(c)** Example mapping event in seqFISH mouse embryo. We show mapping events for 3 Vesalius cost matrices (fncty, fnct, fnt). **(d)** ARI of mapped cell type labels between a reference Slide-seq V2 mouse hippocampus data set and a query Slide-seq V2 mouse hippocampus data set. **(e)** JI of cell interactions of the neighborhood ($k=10$). **(f)** Example mapping event in Slide-seq V2 mouse hippocampus. We show mapping events for 3 Vesalius cost matrices (fncty, fnct, fnt)

3. To me the question how territory/compositional effects contribute towards matching is unclear. The break down in extended figure S1a and S1b seems inconclusive since the “feature” cost already provides a good solution. S1c shows that cell-type is the driving cost. It would be great to get more in-depth on this aspect since the whole point of integrating this approach into Vesalius implies that the point is to leverage this aspect.

We thank the reviewer for bringing this to our attention. We realized that the metrics we had selected to quantitatively score our method did not accurately highlight our algorithms performance. As described above, we updated our scoring method. We applied Vesalius to both synthetic and real data using varying cost matrices and highlight performance changes. We aimed at building on the reviewer’s comment on showing the strengths and weakness of each approach and when one method would be more appropriate than another. Since one the strength of Vesalius is its flexibility in how one might construct their total mapping cost, we added these matrices as part of benchmarking directly instead of showing it in a different figure.

We have included a total of 14 cost matrix combinations (See Figure 2, Figure-3, Supplementary Figure S6 to S9, Supplementary Figure S11 and S12). We show all matrix combinations in the performance plots, but, for clarity, we only show 3 cost matrices for individual mapping events (Figure 2e). Specifically, we selected: *fncty*, *fnct*, and *fnf*. Where *f* stands for cell similarity (feature). *n* stands for niche. *t* stands for territory. *C* stands for cell type composition. *Y* stands for cell type. We selected these 3 cost matrices since they represent either the cost matrices with the highest overall performance (*fncty* and *fnct*) or a matrix combination which does not rely on cell type labels and is directly available through Vesalius analysis alone (*fnf*). For overall performance, we show all cost matrix combinations. The legend for each cost matrix is shown in the body of the figure.

In addition, for real biological data, we investigated metrics related to the cost matrices themselves. First, we computed a coefficient of variation (CV) of each cost matrix. We reasoned that a high CV score would indicate that this cost matrix more is able to discriminate cells much more strongly. For instance, a cost matrix with a correlation spread of 0.2 to 0.6 will be much more able to discriminate cells than a cost matrix with a correlation spread of 0.75 to 0.78. Despite having a higher overall correlation score the second cost matrix has less variation increasing the likelihood of ties or mismatches when other matrices are involved. We also computed a Proportion of Contribution (POC) score which shows how much a specific cost matrix participated in the final cost. Taken together, these metrics show which cost matrices are better suited at discriminating cells and which cost matrices tend to have the highest scores (lowest cost). We highlight these scores for all 3 biological data sets in Supplementary Figure S12.

Changes to the manuscript Results

“Benchmarking mapping performance in real spatial data

Finally, we further investigated the role played by each cost matrix in mapping cells across samples. We first computed a coefficient of variation (CV) on each cost matrix across these 3 biological samples (seqFISH, Stereo-seq, and Slide-seq V2). We reasoned that a higher CV would indicate a higher ability for a cost matrix to discriminate cells during the mapping. We also computed a Proportion of Contribution (POC) which measure the proportion of contribution of each cost matrix to the total cost used for mapping (Supplementary Figure S12). A higher POC is associated with higher JI or correlation. Cell type labels and neighborhood composition show a higher ability to discriminate cells during mapping but participate much less in the total mapping cost. On the contrary, territory and niche similarity tend to have a low CV but participate significantly to the total cost of mapping. “

Supplementary Figures

Figure S12: Cost matrix combinations and their contribution to mapping. We compared the contribution of different cost matrices in 3 biological spatial transcriptomics data sets using CV and POC. **(a)** CV for each combination in seqFISH. The higher the score the more this cost matrix is able to discriminate between cells during mapping. **(b)** POC in seqFISH shows how much each cost matrix contributes to total cost. The higher the contribution the higher the correlation/Jaccard indices were. **(c)** CV for each combination in Slide-seq V2. **(d)** POC in Slide-seq V2 **(e)** CV for each combination in Stereo-seq. **(f)** POC in Stereo-seq

4. Figure 2b on PASTE and GPSA seems inconsistent of how I would consider the ARI to look like in 2a. Do you have an explanation here?

We agree with the reviewers that the scoring of PASTE and GPSA is troublesome. We debated wherever to score these tools or skip them during the quantitative scoring. However, we retained these tools to contrast the task spatial alignment with the task spatial mapping. We believe that these two tasks are distinct but are often confused or assumed to be the same. The issue is that neither tool provides a best matching cell but rather a new set of coordinates for all cells. In most cases, the cells are rotated or distorted but not mapped to the most appropriate spatial location. To provide some sort of score, we computed the K-nearest neighbors between the reference and the query data set and compared the labels of the closets neighbor. We argue that spatial alignment is a distinct task to that of cell mapping which we try to achieve.

In this context, the ARI and JI scores for GPSA and PASTE are not particularly meaningful but were necessary to illustrate the differences in task type. We have added further explanation on this point in the introduction, results, discussion, and methods.

Changes to manuscript

Introduction

“And yet, we have seen the emergence of 3D spatial stacks[8, 10, 11] taken from adjacent tissues and, in parallel, a plethora of tools allowing the alignment of these tissue slices including PASTE[3],GPSA[4], or STalign[12] have been developed. The assumption made by these tools is that adjacent sections share sufficient structural similarity to locally and differentially deform tissue sections to match the neighboring section. Importantly, these methods do not fold, break or scramble the cells present in the tissue rather they displace them by maintaining their relative position to each other.”

Results

“We noticed that spatial alignment tools such as PASTE and GPSA fail to recover the structure of the reference sample from the query sample. The task of mapping cells across samples is a distinct task from spatial alignment and the constraints applied to the samples by PASTE and GPSA are incapable of overcoming these differences.”

Discussion

“Mapping cells across heterogenous samples is a distinct task from spatial alignment of spatial samples. Tools such as PASTE[3], GPSA[4] or STalign[12] make an assumption of structural rigidity which is expected in adjacent tissue samples. While they can locally and differentially deform a query sample to match a target sample, the query sample cannot be broken, folded, or scrambled. However, samples taken from a tumor (even if it is the same tumor) are going to be structurally heterogenous and this heterogeneity will be further amplified when sampling across patients. We demonstrate in our benchmarking (synthetic and real data) that spatial alignment tools (PASTE and GPSA) are incapable of mapping cells between samples in a way that will recover the structure of morphologically complex tissues.”

Methods

“Benchmarking mapping performance

Since PASTE and GPSA do not match cells but rather modify the coordinates of each cell, we computed the nearest cell neighbor between reference and query and compared their cell type labels. While this approach disadvantages PASTE and GPSA, our objective is to demonstrate that context mapping is a fundamentally different task than spatial alignment.

5. Figure 2. It would be great to see what the supposed optimal mapping for the simulated datasets should look like to get a better idea about the point the authors are trying to make.

The optimal mapping would look like the reference data. Our aim is to reconstruct the reference data set from the cells present in the query data sets. The reference shows the optimal mapping while the query represents the “starting” set, the data set we are aiming to transform into the reference. We show “reference” and “query” in Figure 2 (panels in the top left – labeled reference and query respectively) as an example of what we expect to obtain. The same panel and logic are applied to the real data sets where the goal is to map the query cells onto the reference cells (top left panels in Figure 3c/e and Supplementary Figure S11) We have added supplementary figures (Supplementary Figures S1 through S5) demonstrating what each spatial sample should look like. In the case of our benchmarking, we are mapping every sample to every other sample. We exclude self-mapping which results in a total of 132 mapping events.

Figure 2e shows the reference on which we aim to map the cells from the query

6. Details of settings of competing tools is missing. This is particularly problematic, since I think it should be transparent and clear what the differences are. E.g. removing the cell-type information from CytoSpace is in principle ok to make a specific point, but I would object to suppress results with that feature. It is misleading. I rather think the authors should have an open discussion on this feature then (which is also implemented in their own approach).

We agree with the reviewer that we should have been clearer about the settings. We briefly discuss the settings in the method section, but we left the details of parameter selection for the

analysis code available on GitHub. All parameters are explicitly provided for all analysis shown in this manuscript including the generation of the synthetic data. Nevertheless, we have added further details to the methods sections discussing which parameters were used or changed when appropriate.

As for CytoSpace, we originally used true cell type labels instead of providing no labels. However, we were faced with several issues. First, CytoSpace will not run unless all cell type labels are shared between data sets. This made the task of benchmarking synthetic regimes with different cell types present impossible since CytoSpace would simply not run (Raises Value Error written in by CytoSpace developers). We do not expect heterogenous spatial sample to necessarily contain the same cell types. Second, when we used cell type labels and our original scoring method (i.e. ARI of matched cell type labels), we noticed that CytoSpace always obtained a perfect score (ARI = 1) which we found suspicious. CytoSpace also solves a LAP but only on cells of the same type. We decided to omit cell type labels to see if this performance would hold and we found that this was not the case. We did not want to only rely on cell type labels since they do not account for spatial context and might not accurately account for the cell state continuum. While Vesalius can use cell type labels, their use is not as strict as with CytoSpace.

Cell type label cost as computed within Vesalius assigns a score of 1 if the cell type labels are shared and a score of 0 if the cell labels differ. This approach means that while shared labels can improve the mapping, cells will still be mapped according to their cell state potentially overcoming errors in cell type labeling. This also means that missing cell types are allowed within the Vesalius framework since they will simply receive a score of 0 if they do not match to any other cell in the reference data set. These cells can also be filtered out prior to mapping if requested by the user. We have added a discussion of this point to the manuscript.

With that said, we have included CytoSpace *with* and *without* (noLab tag) labels for all our benchmarking when possible. In the synthetic data, we see that CytoSpace with labels always achieves a perfect score in capturing the center cell type as expected (Figure 2, Figure 3, Supplementary Figure S6 through S9, Supplementary Figure S11). When the cell type labels are removed, CytoSpace performs poorly. Vesalius outperforms CytoSpace without labels even when cell types are not included or used to construct cost matrices as it is the case when we use cell, niche and territory similarity.

Interestingly, *even without cell type labels*, Vesalius can outperform CytoSpace *with cell type labels* at capturing the spatial context of cells as shown in the Jaccard Index of cell interaction. We observe the same results across the real biological data indicating that this heavy reliance on cell type labels does not favor accurate mapping of spatial context.

Figure 3a shows the performance of tools and cost matrix combinations in seqFISH data at recovering the identity of cells across samples. We see that CytoSpace obtains a perfect score (ARI) when labels are included but performs poorly when they are not included. This demonstrates CytoSpace's heavy reliance on cell type labels to map across samples.

Figure 3b shows the performance of tools and cost matrix combinations in seqFISH data at recovering the spatial context of cells across samples. In this case, we see that Vesalius can outperform CytoSpace (with and without labels) at recovering the spatial context of cells even when not using cell type labels to construct total cost.

Furthermore, in many instances across the synthetic data and in Stereo-seq, the mapping simply does not occur when using CytoSpace since there are unshared cell type labels between data sets. In dropped regime (Supplementary Figure S7), only 7 out of 132 mapping events occurred successfully.

Changes to manuscript

Discussion

“CytoSpace proved to be an interesting case study on the over-reliance of certain modalities. CytoSpace consistently obtained perfect scores at mapping cell identities across samples in both synthetic and real data. However, this is due to the mechanism by which CytoSpace utilizes cell type labels: Only cells with same cell type label will be mapped to each other. In fact, when cell labels are not shared or removed CytoSpace either fails to run or performs poorly. Even with cell type labels, it does not capture the spatial

context of cells (Figure 2b, Figure 3b/e). While Vesalius can also capitalize on cell type label, it does so with much more flexibility. Cell type labels constitute only part of the total cost matrix and if a cell type label does not match but the other scores (cell, niche, or territory similarity for example) are extremely high, then this mapping event might still occur. This mechanism allows Vesalius to utilize cell labels while still accounting for the cell state continuum and compensating for inaccurate cell type labels.”

Methods

“Benchmarking mapping performance

We contrast our mapping approach to single cell mapping tools, spatial alignment tools, and data integration tools. More specifically, we compared Vesalius to CytoSpace[16], Tangram[16], PASTE[3], GPSA[4], SLAT[5], and Scanorama[1]. It is of note that CytoSpace requires perfect matching of cell type labels between reference and query and only finds a proposed mapping of cells between the same cell type labels. Importantly, if the cell labels are not shared CytoSpace will not run to completion. To demonstrate how this strategy differs from our context specific approach to cell mapping, we provided CytoSpace with true cell labels as well as a single cell label for all cells (noted with the `_noLab` label).

For Vesalius, we ran mapping with 14 cost matrix combinations:

- `feature`
- `niche`
- `territory`
- `composition`
- `feature-niche`
- `feature-niche-cell_type`
- `feature-composition`
- `feature-territory`
- `niche-territory`
- `niche-composition`
- `feature-niche-territory,`
- `feature-niche-composition`
- `feature-niche-composition-territory`
- `feature-niche-composition-territory-cell_type`

Where features represent cell similarity, niche represents niche similarity, territory represents territory similarity, composition represents cell type composition of niche, and cell type represent the cell label score.

To quantitatively assess the performance of cell mapping between samples, we computed an Adjusted Rand Index[6] (ARI) between the cell labels of the query and the reference. We also computed a Jaccard Index between cell labels of the neighborhood of mapped cells to ascertain if we were capturing the spatial and cellular context of each mapped cell. Since PASTE and GPSA do not match cells but rather modify the coordinates of each cell, we computed the nearest cell neighbor between reference and query and compared their cell type labels. While this approach disadvantages PASTE and GPSA, our objective is to demonstrate that context mapping is a fundamentally different task than spatial alignment. Scanorama is

a tool designed to integrate single cell data sets; it only provides an integrated latent space but not best matching cells. As such, we computed the nearest neighbors between the reference data and query data in the integrated latent space to determine best matching cell.

For Vesalius, we ran our mapping using 1000 cells (for synthetic data) or 5000 cells (for real data) across 25 epochs. We set a cell filtering threshold of 0.85 and allowed cell type label filtering if they were provided. We used the log normalized counts to define gene expression signal. Niches were defined using KNN with $k = 6$ for synthetic data and $k = 10$ for real data. In both cases, we select the top 2000 variable genes for analysis.

For CytoSpace, we reformatted synthetic data and real data to fit CytoSpace data requirement. We used 5000 cells as batch size since CytoSpace down-samples based on cell type labels. We ran CytoSpace using their single-cell mode (does not require the deconvolution step to be run).

For Tangram, we loaded data as AnnData objects. We log normalize the data and extract the top 2000 variable genes. We use these variable genes as seed genes for Tangram mapping. We use “rna_count_based” as density prior and ran Tangram for 10 000 iterations.

For PASTE, we loaded data as AnnData objects. Data objects do not need to be normalized prior to alignment. We ran PASTE for 10 000 iterations.

For GPSA, we loaded data as AnnData objects. We ran GPSA for 50 iterations as recommended by authors. The learning rate for the Adam optimizer was set at 0.01 as recommended by authors.

For SLAT, we loaded data as AnnData objects. SLAT does not require data to be normalized prior to mapping. We selected neighborhoods using K-NN ($k=6$ for synthetic data and $k = 10$ for real data). We ran SLAT with 5 Graph Convolutional layers across 25 epochs. Parameters were modified in accordance to developer recommendations.

For Scanorama, we load data as AnnData objects and run Scanorama with default parameters.

For all benchmarking, all code and parameters used in this analysis is available in the dedicated GitHub.”

7. A discussion on the compute requirements compared also to the other approaches would be warranted, since current datasets are increasing in complexity.

We included a benchmarking for computational performance by varying the number of cells used during mapping (100 to 200k). For this task, we aimed at making the performance as fair as

possible as such we assumed that a user would run a full analysis pipeline including data loading, preprocessing, mapping, and exporting results on a single CPU.

We made this assumption since all tools have different input requirements and different downstream tasks. Vesalius provides the widest variety of potential downstream task and as such has the most requirement in terms of pre-processing. This strategy puts our tool at a disadvantage, but it was the least objectionable way to benchmark performance.

We also aimed to avoid any variation in language specific methods of measuring run time and memory usage. We measured run time and memory (Resident Set Size – peak RAM usage) for each tool using custom bash scripts (Available on the dedicated GitHub page) that measured these metrics across 3 submitted jobs. We set hard computational limits: a maximum of 480GB of RAM and maximum run time of 12 hours. If the job exceeded these limits, the run was cut short. Despite its higher pre-processing requirements, Vesalius strikes a balance between mapping performance, computational performance and interpretability. We show computational performance in Supplementary Figure S10. We also include a comparison of run time and memory usage across cost matrix combinations for 5000 and 10 000 cells.

Run time can be significantly improved by making use of multiple cores which is already part of the package. The high memory requirement stems from computing large cost matrices (200k by 200k) which could be done in batches (batches are already used when using multiple cores) and using on disk computations. We will consider using on disk computations for further iterations of the package. It is of note that Vesalius is the only R package tested and while newer version of R with vectorized code and an Rcpp backend (which we use) can be performant, the language still lags behind python in terms of run time and memory performance. Vesalius can be favored by biologist more comfortable in R despite lower performance.

Changes to the Manuscript

Results

“Finally, we highlight computational performance (run time and memory usage) when increasing the number of cells from 100 to 200 000 (Supplementary Figure S10) in a synthetic circle regime. For each tool, we computed the mean run time and peak RAM usage (Resident Set Size) across 3 runs for each number of cells (see methods). We also include a comparison of run time and memory usage across cost matrix combinations for 5000 and 10 000 cells.”

Methods

“In addition to these data sets, we also generated 3 samples of the circle regime with varying number of cells (100, 500, 1000, 2500, 5000, 10000, 20000, 50000, 100000, 200000). These data sets were used for computational performance benchmarking (see below).”

“Computational performance benchmarking

We included a benchmarking for computational performance by varying the number of cells used during mapping (100, 500, 1000, 2500, 5000, 10000, 20000, 50000, 100000, 200000) in a circle synthetic regime.

For this task, we aimed at making the performance as fair as possible as such we assumed that a user would run a full analysis pipeline including data loading, preprocessing, and mapping on a single CPU. We made this assumption since all tools have different input requirements, different

downstream tasks, and different optimization options. Vesalius provides the widest variety of potential downstream task and as such has the most requirement in terms of pre-processing.

We also aimed to avoid any variation in language specific methods of measuring run time and memory usage. We measured run time and memory usage (Resident Set Size – peak RAM usage) for each tool using custom bash scripts (Available on the dedicated GitHub page) that measured these metrics across 3 submitted jobs. We set hard computational limits: a maximum of 480GB of RAM and maximum run time of 12 hours. If the job exceeded these limits, the run was cut short.

We compared run time and memory usage across cost matrix combinations for 5000 and 10 000 cells. We used the same 14 matrix combinations described above and used the same performance measuring strategy as described above.

”

Figure-S10: Mean computational performance across 3 runs. Each missing tile signifies that a run exceeds either max run time (12 hours) or max allocated memory (480GB). (a) Mean Run time across all tools with increasing number of cells (b) Mean peak memory usage across all tools with increasing number of cells. (c) Mean Run time across Vesalius cost matrix combinations for 5000 and 1000 cells. (d) Mean peak memory usage across Vesalius cost matrix combinations for 5000 and 1000 cells.

8. There are different types of tasks demonstrated here. In principle, Vesalius can handle them, but it does so by switching on/off scenarios that influence the cost function. While I appreciate this flexible platform, from an evaluation perspective one should appropriately benchmark with the according competing tools.

We appreciate the reviewer bringing this to our attention. Our original intention was to demonstrate flexibility in a variety of biological and analysis scenarios. We hoped to show that the analysis can be tailored to the question at hand. In addition, some information might not always be available depending on the data sets used. For instance, accurate cell types might not be available for cancer data sets. Our aim was to demonstrate how Vesalius can maximize the utilization of the information that is available and even be tailored to the researcher's interests. Nevertheless, we performed all the benchmarking using different combinations of cost matrices across synthetic and real data to demonstrate their effect and to increase benchmarking consistency.

We have included a total of 14 cost matrix combinations (See Figure 2, Figure-3, Supplementary Figure S6 to S9, Supplementary Figure S11 to and S12). We show all matrix combinations in the performance plots, but, for clarity, we only show 3 cost matrices for individual mapping events (Figure 2e). Specifically, we selected: *fncty*, *fnct*, and *fmt*. Where *f* stands for cell similarity (feature). *n* stand for niche. *t* stands for territory. *C* stands for cell type composition. *Y* stands for cell type. We selected these 3 cost matrices since they represent either the cost matrices with the highest overall performance (*fncty* and *fnct*) or a matrix combination which does not rely on cell type labels and is directly available through Vesalius analysis alone (*fmt*). For overall performance, we show all cost matrix combinations. The legend for each cost matrix is shown in the body of the figure.

In the case of the analysis presented in Figure 5, Figure 6 and Supplementary Figure S15, we tailored the analysis based on what question we were trying to answer. Figure 5 shows mapping of cells across regenerative time and developmental time. We only used cell similarity, niche similarity and territory similarity in this analysis since we aimed at finding differences in large scale tissue structures such as the zone of injury (Figure 5a-c) in axolotl brain generation data. The same logic was applied to the mouse development data shown in Figure 5d-g. Furthermore, since this analysis pertains to mapping cells through space and time, the cell type labels do not match even though they are comparable cell types. Our analysis shows that despite these discrepancies Vesalius can still provide valuable insight into the data. Interestingly, the flexibility of Vesalius would allow the addition of fate change probabilities as a cost matrix.

In Figure 6, we utilized all available information provided in the IMC data which includes cell similarity, niche similarity, territory similarity, cell type composition, and cell type labels. We also maximized the use of available information when mapping 2 prostate cancer tumor samples in Supplementary Figure S15.

We hope that the benchmarking appropriately demonstrates the different scenarios and how they perform while the analysis presented in the rest of the manuscript highlight how the flexibility offered by Vesalius is an asset to researchers. The ability to add custom cost matrices would in theory allow a researcher to add lineage fate or specific gene marker expression as a potential cost matrix for example.

9. I did find the manuscript difficult to read and I think this is due to the structure. Please feel free to ignore this comment if the authors feel strongly against this, however from a paper reading perspective, organising it based on the tools features rather than by technology may make it easier to read and follow (.e.g using LAP with default features, using LAP with default feature and optional feature etc)

As described above, our original intent was to organize the manuscript by biological question and analysis scenario which we hoped would highlight the features of the package. Following the reviewer's suggestion, we have re-organized the paper to make the structure clearer. We first show benchmarking performance in synthetic data sets using various cost matrices and comparing performance to other tools. We perform the same benchmarking in real data still using various cost matrices and comparing to other tools. Next, we demonstrate Vesalius's ability to answer biological questions in various scenarios where the choice of cost matrix will depend on the information available to the user. We hope that this structure more clearly demonstrates our intent when developing this package.

10. Formatting is inconsistent at some places (e.g.: Supplemental Figure S2 vs Supplemental Figure 3)

We updated the text for consistency.

11. Some grammatical mistakes here and there. I suggest to run it through an according grammar checker of your choice.

We checked grammar and made changes accordingly.

12. Vesalius citation has mark-down elements visible

We remove markdown elements.

13. Wish for more mathematical formulation of the method section (e.g.: concrete cost function). One can piece it together based on the information provide, but it would help with clarify of more technical people.

We have added explicit mathematical formulation when needed.

Changes to the manuscript

Methods

"Mapping cells across samples

To map cells across samples, Vesalius employs the Jonker-Volgenant algorithm ($O(n^3)$) [16], a variant of the LAP. The algorithm is to map "workers" to "task" while minimizing the overall cost. To allow many-to-

many matching of cells, we implemented the solver in a divide-and-conquer framework which also provides a speed up for large batches through parallelization.

Given a requested batch size B , we randomly sample cells from the reference set R and the query set Q . We select a subset $R_B \subset R$ and $Q_B \subset Q$ such that $|R_B| = |Q_B| = B$. Once a cell is selected, it is removed from the sampling pool to ensure all cells are eventually selected. If the number of cells in one dataset is smaller than in the other, padding is applied. Without loss of generality, assume $|R| < |Q|$. In this case, we create a duplicated set R' and concatenate it with R , forming $R^* = R \cup R'$, where $|R^*| \geq B$ or matches $|Q|$, whichever is smaller. The Linear Assignment Problem (LAP) solver is then applied to each batch to optimize the mapping between the sampled reference and query cells. Given a cost matrix C for a batch, the optimal assignment M^* is found by minimizing the total cost:

$$M^* = \arg \min \sum_{(i,j) \in M} C_{ij}$$

If a one-to-one matching is required, duplicate matches can be filtered by retaining only the pairs with the lowest overall cost.

Since the mapping depends on the cells selected during sampling, the batching and solving steps can be applied across multiple epochs. At each epoch t , new random batches $R_B^{(t)} \subset R$ and $Q_B^{(t)} \subset Q$ are selected and mapped using the LAP solver. Let $M^{(t)}$ represent the set of matched cell pairs at epoch t . The newly matched cell pairs are compared to those from previous epochs, and a pair (i, j) is retained if it results in a lower matching cost:

$$C_{ij}^{(t)} < C_{ij}^{(t-1)}$$

where $C_{ij}^{(t)}$ is the cost associated with matching cell i from R to cell j from Q at epoch t .

The final output is a many-to-many mapping of cells across samples, along with the mapping cost, epoch information, and mapping scores for each cell pair.

Computing cost

To solve the LAP, we compute a total cost matrix constructed from comparing cells through a variety of biological features. The default features used by Vesalius are the following:

- Cell similarity (f): compares the gene expression of cells between each sample.
- Niche similarity (n): compares the gene expression of cell niches between each sample.

Optionally:

- Territory similarity (t): compares the gene expression of territories between each sample.
- Cell label (y): checks whether both cells share the same cell type label.
- Composition similarity (c): compares the cell type composition of cell niches between each sample.

- Custom similarity: compares cells between samples using user defined similarities.

To generalize the formula for an arbitrary number of similarity matrices, we define a set n of similarity matrices $S^{(k)}$ for $k = 1, \dots, n$. The final cost matrix C is then computed as the sum of the reciprocals of all provided similarity matrices:

$$C_{ij} = \sum_{k=1}^n (1 - S_{ij}^{(k)})$$

where $S_{ij}^{(k)}$ represents the similarity score between query cell i and reference cell j in the k -th similarity matrix. This formulation allows the incorporation of any number of similarity metrics.

In the case of continuous features (cell, niche, and territory), we first extract a gene expression signal. The signal represents a gene expression profile (normalized or raw) of all genes, highly variable genes, or a custom gene set. Alternatively, embeddings values (PCA, UMAP, LSI, NMF) can also be used to define the gene expression signal. By default, and across all analysis presented here, we used the highly variable genes. Since genes might not perfectly overlap between data sets, we select the genes which lie at the intersection of gene expression signals for each data set.

We calculate a Pearson's correlation coefficient between expression signals of each potential cell pair:

$$r = \frac{\sum(x_i - \bar{x})(y_i - \bar{y})}{\sqrt{\sum(x_i - \bar{x})^2 \sum(y_i - \bar{y})^2}}$$

where x_i and y_i are the expression of genes i in sample x and y respectively. \bar{x} and \bar{y} are the mean expression of cell x and cell y .

For categorical features such as niche composition, we compute a frequency aware Jaccard index:

$$J(A, B) = \frac{|A \cap B|}{|A \cup B|}$$

We assign a unique label to duplicated cell type labels. If cell type A is present more than once, the first instance the label will be A , the second instance $A.1$, and so on.

Similarity matrices which are computed using Pearson's correlation coefficient or a Jaccard Index can be filtered by a user defined threshold prior to mapping. For all cells in the query, we check if it has at least one match higher than the threshold.

We define an indicator function for the label agreement between a reference cell j and a query cell i . Let L_i and L_j be the labels of the query cell i and reference cell j , respectively. Then, we define:

$$I_{ij} = \begin{cases} 1, & \text{if } L_i = L_j \\ 0, & \text{if } L_i \neq L_j \end{cases}$$

This function assigns a value of 1 if the labels match and 0 otherwise. A query cell i that satisfies the following condition:

$$\sum_i I_{ij} = 0$$

Is assumed to not have any shared cell type labels with the reference data and can be removed upon user request prior to mapping.

Extracting Niches

For each cell, we define a niche as the spatially neighboring cells using the following methods:

- KNN: K-Nearest Neighbors
- Graph: using Voronoi Tessellation, we define a neighborhood graph and select niches by considering graph depth from a center cell. e.g. direct neighbors will have a graph depth of 1.
- Radius: All cells within a certain radius of a center cell are defined as this cell's niche.

In the context of niche similarity, let X_c be the gene expression profile of cell c , and $N(c)$ be the set of cells in the neighborhood of cell c .

1. **Mean gene expression profile of the neighborhood for cell c :**

$$\bar{X}_{N(c)} = \frac{1}{|N(c)|} \sum_{c' \in N(c)} X_{c'}$$

where $|N(c)|$ is the number of cells in the neighborhood of c , and $X_{c'}$ is the gene expression profile of cell c' .

2. **Gene expression profile of cell c becomes the average expression of its neighborhood:**

$$X_c = \bar{X}_{N(c)}$$

Extracting Territories

Originally, Vesalius was designed to detect tissue territories from spatial transcriptomics data using image processing techniques. We have since improved the territory detection abilities. In addition to a wider range of dimensionality reduction techniques (PCA, UMAP, Latent Semantic Indexing (LSI), Non-Negative Matrix Factorization (NMF), custom) and normalization methods (log normalization, SCTransform, Term-Frequency Inverse Document Frequency (TF-IDF), custom), we have adapted the image processing steps of Vesalius to function on image tensors instead of RGB formatted images. This upgrade allows us to apply the image processing steps to all latent space dimensions simultaneously and in turn improve our ability to recover tissue territories.

In summary, spatial transcriptomics data is converted into a grey scale image stack. We normalized counts and extract highly variable features. With these features, we compute a reduced dimension latent space for each cell. In parallel, we expand punctual coordinates using Voronoi tessellation and rasterize the resulting tiles to produce a grey scale image for each latent space dimension. This image stack can be processed using a variety of image processing techniques such as histogram equalization, image smoothing, and image segmentation. Color segments can be further divided to distinguish spatially distinct color segments. The details of this process are discussed in our previous publication[17].

The territories isolated by Vesalius are used to compare the expression of cells across larger scale spatial domains. We compare the average expression of the territory in which a cell finds itself across samples.

Let T be a territory defined by Vesalius. Let X_c be the gene expression profile of cell $c \in T$, and $T(c)$ be the set of cells in Territory T .

- 2. Mean gene expression profile of the neighborhood for cell c :**

$$\bar{X}_{N(c)} = \frac{1}{|T(c)|} \sum_{c' \in T(c)} X_{c'}$$

where $|T(c)|$ is the number of cells territory T , and $X_{c'}$ is the gene expression profile of cell c' .

- 3. Gene expression profile of cell c becomes the average expression of its neighborhood:**

$$X_c = \bar{X}_{T(c)}$$

14. Introduction generally points out existing tools, but does not put it in context of the approach of the authors.

We added tool contextualization to the introduction.

Changes to manuscript

Introduction

“With the increasing availability and breadth of ST technologies, it comes to no surprise that they have become part of the biomedical research arsenal. It is still difficult and expensive to obtain high quality spatial data especially from human patients. Therefore, there is a need to maximize the use of the spatial data deposited in the public domain. However, data sets that are already available in a biological condition of interest might have been produced using different methodologies or technologies. Moreover, in a clinical setting where a researcher might be interested in following the evolution of cells and their interactions across disease progression or as a response to treatment, it is difficult to expect tissues to share similar structures. And yet, we have seen the emergence of 3D spatial stacks[8, 10, 11] taken from adjacent tissues and, in parallel, a plethora of tools allowing the alignment of these tissue slices including PASTE[3],GPSA[4], or STalign[12] have been developed. The assumption made by these tools is that adjacent sections share sufficient structural similarity to locally and differentially deform tissue sections to match the neighboring section. Importantly, these methods do not fold, break or scramble the cells present in the tissue rather they displace them by maintaining their relative position to each other. But the irregular nature of tumors across patients and conditions restricts their use to highly structured tissues. As such, the question remains: how do we compare the spatial context of individual cells when tissue structures do not match across samples or where sequential sampling is simply impossible?”

Mapping single cell data to a spatial assay based on the transcriptome may suggest an alternative solution to align two or more ST assays based on their transcriptome[2, 18]. For instance, Tangram uses a deep-learning approach to match sc/snRNA-seq to their estimated spatial location by maximizing the spatial correlation between scRNA and spatial data[18]. CytoSpace solves a linear assignment problem (LAP) to match single cells to spatial locations by accounting for cell type proportions in low resolution Visium spots[2]. In conjunction with single cell labels, CytoSpace can also map to high-resolution spatial data. Along the same lines, methods designed to integrate single cell data such as Scanorama[1] can find the closest related cells in their integrated latent space. However, data mapping or integration without spatial and biological context can lose the information related to tissue or cellular microenvironment. Algorithms such as Spatial-linked alignment tool (SLAT) provide a solution to spatial mapping by adopting a graph adversarial matching strategy[5]. Interestingly, SLAT provides an integrated latent space making it a spatially aware data integration method. SLAT represents cells as neighborhood graphs and applies a graph convolution neural network to generate its embeddings. This strategy can reduce both context flexibility and interpretability. While SLAT considers the spatial context of cells across samples, it has not been tested systematically for tissues whose tissue structure is highly dissimilar. More importantly, current mapping strategies cannot be applied to systemic, flexible, and interpretable comparison of hundreds of samples. “

15. Figure 5 axis labels not readable

It is now Figure 6. Due to space constraints, we were unable to increase the size of the labels for 100 patients. The labels represent the sample IDs. In Figure 6a and b, the x axis labels are the best matching pairs. In figure 5c, the axes are all sample IDs. Finally, in Figure 6d, the x-axis label are the IDs of data sets belonging to a cluster. For clarity, we have included best matches and cluster belonging to Supplementary Table 4 and specified this in text.

Changes to text

Results

“Labels and clustering results are also available in Supplementary Table 4”

References

1. Hie, B., B. Bryson, and B. Berger, *Efficient integration of heterogeneous single-cell transcriptomes using Scanorama*. Nature Biotechnology, 2019. **37**(6): p. 685-691.
2. Vahid, M.R., et al., *High-resolution alignment of single-cell and spatial transcriptomes with CytoSPACE*. Nature Biotechnology, 2023. **41**(11): p. 1543-1548.
3. Zeira, R., et al., *Alignment and integration of spatial transcriptomics data*. Nat Methods, 2022. **19**(5): p. 567-575.
4. Jones, A., et al., *Alignment of spatial genomics data using deep Gaussian processes*. Nature Methods, 2023. **20**(9): p. 1379-1387.

5. Xia, C.-R., et al., *Spatial-linked alignment tool (SLAT) for aligning heterogenous slices*. Nature Communications, 2023. **14**(1).
6. Rand, W.M., *Objective Criteria for the Evaluation of Clustering Methods*. Journal of the American Statistical Association, 1971. **66**(336): p. 846-850.
7. Lohoff, T., et al., *Integration of spatial and single-cell transcriptomic data elucidates mouse organogenesis*. Nature Biotechnology 2021 40:1, 2021. **40**(1): p. 74-85.
8. Chen, A., et al., *Title: Large field of view-spatially resolved transcriptomics at nanoscale resolution Short title: DNA nanoball stereo-sequencing*. bioRxiv, 2021: p. 2021.01.17.427004-2021.01.17.427004.
9. Stickels, R.R., et al., *Highly sensitive spatial transcriptomics at near-cellular resolution with Slide-seqV2*. Nature Biotechnology, 2021. **39**(3): p. 313-319.
10. Maynard, K.R., et al., *Transcriptome-scale spatial gene expression in the human dorsolateral prefrontal cortex*. 2021. **24**(3).
11. Wei, X., et al., *Single-cell Stereo-seq reveals induced progenitor cells involved in axolotl brain regeneration*. Science, 2022. **377**(6610): p. eabp9444.
12. Clifton, K., et al., *STalign: Alignment of spatial transcriptomics data using diffeomorphic metric mapping*. Nature Communications, 2023. **14**(1).
13. Hirz, T., et al., *Dissecting the immune suppressive human prostate tumor microenvironment via integrated single-cell and spatial transcriptomic analyses*. Nature Communications, 2023. **14**(1): p. 663.
14. Cable, D.M., et al., *Robust decomposition of cell type mixtures in spatial transcriptomics*. Nature Biotechnology, 2021: p. 1-10.
15. Saunders, A., et al., *Molecular diversity and specializations among the cells of the adult mouse brain*. Cell, 2018. **174**(4): p. 1015-1030. e16.
16. Jonker, R. and A. Volgenant, *A shortest augmenting path algorithm for dense and sparse linear assignment problems*. Computing, 1987. **38**(4): p. 325-340.
17. Martin, P.C.N., et al., *Vesalius: high-resolution in silico anatomization of spatial transcriptomic data using image analysis*. Molecular Systems Biology, 2022. **18**(9).
18. Biancalani, T., et al., *Deep learning and alignment of spatially-resolved whole transcriptomes of single cells in the mouse brain with Tangram*. bioRxiv, 2020: p. 2020.08.29.272831-2020.08.29.272831.

We thank the reviewers for their positive remarks and their follow up questions. We have addressed the reviewers concerns below. For clarity, we provide our response to reviewer comments in a color-coded manner. Blue represents the reviewer's concern, in black our response (if any) and in red the changes we made to address the comments. We also include the text changes and the newly generated figures (when appropriate). The text changes within the manuscript itself are highlighted in red.

Reviewer #1 (Remarks to the Author)

We thank the authors for the well organized and thorough explanations addressing our comments. The revised introduction clarifies the utility of this approach compared to rigid and diffeomorphic alignment algorithms that assume structural coherence. The new runtime benchmark is also greatly appreciated and helps set expectations for reasonable runtimes despite the big(O) complexity. We will look forward to trying out this tool on our data.

Our only major remaining question is: in the revised text, we now better appreciate that Versalius is incorporating a local spatial context to create a joint embedding. Other approaches also seek to create a joint embedding but doesn't take such spatial context into consideration as well explained by the authors. However, what is modulates the size of this spatial context that Versalius takes into consideration? From the methods, it looks like either a KNN, Graph, or Radius distance is used. Do the authors have recommendations/guidance on how to determine an appropriate niche size?

Particularly since the benchmarking is later done assuming a $k=6$ neighborhood size, we're wondering how these hyper parameters interact. For example, if we choose a niche size <6 , then will benchmarks assuming a $k=6$ neighborhood size in the benchmark suggest poorer performance?

The point brought up by the reviewer is a crucial one when dealing with local spatial context and cellular niches. Many factors can influence this choice including cell size and density in choosing niche size. This remains an open question as the size of the niche will largely depend on the question at hand.

With that said, in the context of cell mapping with Vesalius, we recommend using the graph method with a depth of 1 as a starting point. This method creates a spatial graph using Delauney Triangulation ensuring that neighboring cells are in direct contact with the center cell.

A depth of 1 means that we will only use the first "ring" around the center cell. This approach provides a solid approach to capture the local spatial context of a cell – especially when using niche similarity - and recovering complex tissue structures. While we recommend the use the graph method, we urge any user to carefully consider their question and tailor their analysis (and parameter selection) to the question at hand.

There are instances where we opted for alternative method such as K-NN or Radius. For benchmarking, where the aim is robust scoring of performance, we used K-NN for 3 reasons:

1. SLAT requires to define a neighborhood size using K-NN. To make the benchmarking as fair as possible we used the same method to define neighborhood.
2. We needed to define a spatial context which we could use for scoring the performance of each framework. K-NN would ensure a constant neighborhood size and remove any issue with neighborhood size normalization.

3. K=6 is a reasonable estimation as the number of neighboring cells when considering a graph depth of 1.

We expect poorer performance for $k < 6$ as the information used decreases. To directly answer the reviewer's question, we performed another round of benchmarking on a synthetic data set with varying neighborhood sizes ($k = 2 \dots 14$) using all cost matrices. Despite changing the size of the neighborhood, Vesalius is still able to accurately capture the spatial context of cells. As expected, the performance is slightly reduced when using $k < 6$ and showed its peak when $k=7$. These results demonstrate that Vesalius is robust to parameter selection.

We used Radius when mapping across resolutions (VisiumHD onto Visium) since this was the only appropriate way to compare niches given difference in cellular densities which can occur in real tissues. With a fixed radius, VisiumHD niches may contain different number of cells which more closely mimics what we observe in Visium data and still provides comparable spatial niches.

Methods

In most cases, we recommend using *Graph* since this method best captures the biological realities of cellular organization. We urge any user to carefully consider their question and tailor their analysis (and parameter selection) to the question at hand.

Likewise, hearkening back to the title of the paper which focuses on multi-scale integration, how does Vesalius parameters prioritize local versus global niche structures? For example, there may be some samples that are compositionally similar on a fine scale but rather different on a coarser/global scale, particularly in the context of cancer. Or alternatively, there may be diseased tissues of structurally stereotypic organs like brains that are quite similar on a coarser/global scale to healthy tissues but locally very different. Is Vesalius able to guide users to the appropriate scale of similarity/differences?

We thank the reviewer for this observation and question. Interestingly, it is for these reasons that we designed Vesalius with flexibility in mind. The question here boils down to which cost matrix are the most appropriate to answer a specific question.

As described in the methods section, cost matrices using a Pearson Correlation Coefficient (cell, niche, territory) will have score ranging from -1 to 1. Cell composition of niches uses a Jaccard Index which provides scores between 0 and 1. And finally cell type labels provide cost matrices composed of either 0 or 1. The total cost is summed across all selected cost matrices using element wise summation. Since all cost matrices can have a maximum score of 1, they have all have the same weight in the total cost from a computational perspective. In other words, our method does not apply explicit weighting to any cost matrix.

A key advantage of our method is its robustness to mismatches. For instance, if two cells have identical cell type labels but low similarity across other features, they may not be mapped to one another. Conversely, a cell may be aligned with a different cell type if other similarity metrics—such as expression correlation or niche composition—indicate a stronger match. This flexibility addresses a key limitation of tools like CytoSpace, which restrict mappings strictly to identical cell type labels.

In this framework, global mismatches (e.g., territory-level differences) reduce the matching score at the territory scale, while strong local similarity can still drive accurate mappings. Conversely, if local niche composition differs while cells remain within the same global domain, local scores may be lower, but higher territory-level similarity compensates in the final matching decision.

In essence, Vesalius is a powerful and versatile tool to map cell across samples which empowers the use to tailor their analysis to what makes the most sense for them.

Building remarks from other reviewers, we have added some recommendations in the discussion.

Discussion

A key aspect to consider when using our approach is which cost matrix to utilize to obtain optimal cell mapping. In Supplementary Figure S12, we demonstrate how each cost matrix may contribute to discriminating between cells and participates in the total cost. Discrete and categorical cost matrices (cell composition and cell type labels) tend to have a stronger effect on optimal mapping as seen in Figure 2, Figure 3, and Supplementary Figure S12. The assumption is that cell type labels (and by extension niche composition) accurately represent prior biological knowledge. Our algorithm uses this prior biological knowledge to its advantage without requiring explicit weighting

of each cost matrix. Across all our benchmarking, we observed that using all available information (cell, niche, territory similarities, niche composition, and cell type) provides the best combination at recovering center cell identity and cellular context. With that said, if cell type labels are not available or are not deemed appropriate (e.g. different nomenclature), using cell and niche similarity can still provide highly accurate mapping of cells across samples. We recommend including territory information since cells can express global context markers as we previously demonstrated¹. Nevertheless, it is crucial to tailor any analysis to the question at hand.

I was asked to comment on responses to Reviewer #3's comments.

Regarding Comment 1-2: The authors now provide more thorough benchmarks with real ST data, addressing these previously raised concerns.

Regarding Comment 3: The reviewer's previous comment highlights a point of confusion as to how territory/compositional effects contribute to the cost function, particularly since cell-type composition appears to be driving this cost function. While the revision includes equations defining the cost matrix and quantifying niches, territories, etc, it remains unclear how these various quantifications are weighted.

While all cost matrix combinations are shown, the interpretation remains unclear. A discussion on how to best use `fcty`, `fct`, `fnt`, etc modes of `Vesalius` is needed.

We thank the reviewer for taking the time to go through these comments.

To summarize, our cost matrices are not weighted however discrete inputs (cell type and niche composition) which are assumed to be accurate prior biological knowledge will have a stronger effect. Our algorithm uses this prior biological knowledge to its advantage without requiring explicit weighting of each cost matrix. Across all our benchmarking, we observed that using all available information (cell, niche, territory similarities, niche composition, and cell type) provides the best combination at recovering center cell identity and cellular context across all data sets. It consistently performed well across test cases making it the most appropriate cost matrix combination. With that said, if cell type labels are not available or are not deemed appropriate (e.g. different nomenclature), using cell and niche similarity provides highly accurate mapping of cells across samples (e.g. Figure S11). We recommend including territory information since cells express global context markers as we previously demonstrated¹.

The difference in effect stems from the type of information that is being used in the construction of the cost matrix. In the case of cell type labels or niche composition, we are using discrete/categorical data based on prior biological knowledge assumed to be true. This means that scores will “jump” more drastically between possible states in comparison with continuous metrics such as transcriptional similarity. To illustrate, if a cell is surrounded by 6 cells and we accurately recover the cell type label of the center cell and its niche composition this will result in a perfect score for both metrics. This translates into a zero cost. Despite being able to capture these labels accurately, it is unlikely that continuous metrics measuring a Pearson's correlation coefficient will yield a perfect score. By assuming that the prior biological is accurate, our algorithm can increase the accuracy of mapping without needing to explicitly weight cost matrices.

A key advantage of our method is its robustness to mismatches. For instance, if two cells have identical cell type labels but low similarity across other features, they may not be mapped to one another. Conversely, a cell may be aligned with a different cell type if other similarity metrics—such as expression correlation or niche composition—indicate a stronger match. This flexibility addresses a key limitation of tools like CytoSpace, which restrict mappings strictly to identical cell type labels.

We have added a section to the discussion to provide some guidance to potential users as well as some further explanations of our approach in the methods section.

Discussion

A key aspect to consider when using our approach is which cost matrix to utilize to obtain optimal cell mapping. In Supplementary Figure S12, we demonstrate how each cost matrix may contribute to discriminating between cells and participates in the total cost. Discrete and categorical cost matrices (cell composition and cell type labels) tend to have a stronger effect on optimal mapping as seen in Figure 2, Figure 3, and Supplementary Figure S12. The assumption is that cell type labels (and by extension niche composition) accurately represent prior biological knowledge. Our algorithm uses this prior biological knowledge to its advantage without requiring explicit weighting of each cost matrix. Across all our benchmarking, we observed that using all available information (cell, niche, territory similarities, niche composition, and cell type) provides the best combination at recovering center cell identity and cellular context. With that said, if cell type labels are not available or are not deemed appropriate (e.g. different nomenclature), using cell and niche similarity can still provide highly accurate mapping of cells across samples. We recommend including territory information since cells can express global context markers as we previously demonstrated¹. Nevertheless, it is crucial to tailor any analysis to the question at hand.

Methods

Importantly, cost matrices are not weighted. Cost matrices using discrete/categorical inputs (niche composition and cell type labels) leverage prior biological knowledge and will yield a stronger discriminative ability without needing to explicitly be weighted.

Regarding Comment 4: The revised text now clarifies the distinction spatial alignment approaches versus Vesalius

Regarding Comment 5: The revised figure addresses this previous concern

Regarding Comment 6: The reviewer was looking for the default settings used for competing tools to ensure reproducibility. It would be appropriate to include version number of default parameters in methods.

For each tool, we modified default parameters to ensure that all tools were running under similar conditions. We specify which parameters were changed in the methods section. In addition we

have added tool version as suggested by the reviewer. Please note that we use the official name in text and specify the package name and version if it differs from the official tool name.

Finally, we have added `ym/` files to the analysis GitHub which also contains the entire analysis with selected parameters. Each tool contains a full environment dependency listing (https://github.com/WonLab-CS/Vesalius_analysis/tree/main/benchmarking)

Methods

For Vesalius (==2.0.0), we ran our mapping using 1000 cells (for synthetic data) or 5000 cells (for real data) across 25 epochs. We set a cell filtering threshold of 0.9 and allowed cell type label filtering if they were provided. We used the log normalized counts to define gene expression signal. Niches were defined using KNN with $k = 6$ for synthetic data and $k = 10$ for real data. In both cases, we select the top 2000 variable genes for analysis.

For CytoSpace (==1.0.6a0), we reformatted synthetic data and real data to fit CytoSpace data requirement. We used 5000 cells as batch size since CytoSpace down-samples based on cell type labels. We ran CytoSpace using their single-cell mode (does not require the deconvolution step to be run).

For Tangram (tangram-sc==1.0.4), we loaded data as AnnData objects. We log normalize the data and extract the top 2000 variable genes. We use these variable genes as seed genes for Tangram mapping. We use “rna_count_based” as density prior and ran Tangram for 10 000 iterations.

For PASTE (paste-bio==1.4.0), we loaded data as AnnData objects. Data objects do not need to be normalized prior to alignment. We ran PASTE for 10 000 iterations.

For GPSA (==0.8), we loaded data as AnnData objects. We ran GPSA for 50 iterations as recommended by the authors. The learning rate for the Adam optimizer was set at 0.01 as recommended by the authors.

For SLAT (scslat==0.2.1), we loaded data as AnnData objects. SLAT does not require data to be normalized prior to mapping. We selected neighborhoods using K-NN ($k=6$ for synthetic data and $k = 10$ for real data). We ran SLAT with 5 Graph Convolutional layers across 25 epochs. Parameters were modified in accordance to developer recommendations.

For Scanorama (==1.7.4), we loaded data as AnnData objects and run Scanorama with default parameters.

For all benchmarking, all code, parameters, and dependencies lists used in this analysis are available in the dedicated GitHub.

Regarding Comment 7: The new runtime benchmarks address this previous concern

Regarding Comment 8: This relates to Comment 3 and remains to be thoroughly addressed

We thank the reviewer for looking through the comments. Since this comment relates to comment 3, please see our response under comment 3.

All other minor comments have been addressed

(Remarks on code availability):

Looks reasonably well documented though we have not tried to install or reproduce the results.

We confirm the website is now functional and vignettes are compiled.

Reviewer #2 (Remarks to the Author)

The additions made to the paper are appropriate, clarifying some concerns raised by reviewers. In particular, the addition of benchmarking has greatly clarified the benefits compared to other similar tools e.g. matching cells from non-adjacent tissue, and inclusion of spatial context to the model. Of note, they have included prostate cancer samples to test the viability of Vesalius in highly heterogeneous spatial datasets, successfully demonstrating the benefit this tool would provide for spatial cancer research, which often works with multiple sections of FFPE tissue and has difficulty cell typing across tissue. Overall, the paper provides a spatial specific tool that would be highly beneficial in the current spatial biology landscape.

We thank the reviewer for their constructive input on our manuscript.

References

- 1 Martin, P. C. N., Kim, H., Lövkvist, C., Hong, B. W. & Won, K. J. Vesalius: high-resolution in silico anatomization of spatial transcriptomic data using image analysis. *Molecular Systems Biology* **18** (2022). <https://doi.org:10.15252/msb.202211080>